# Task, person, and experiential characteristics drive the transfer of learning
Kyle J. LaFollette[1,2] ✉, David J. Frank ●[3], Alexander P. Burgoyne ●[4,5] & Brooke N. Macnamara ●[1,6]

The ability to transfer skills is critical for complex performance. However, performance in complex environments is often examined within single levels of analysis, neglecting interactions among characteristics of the task, person, and experience. Here, we examine how intervention-level factors (task consistency, stress), between-person differences (emotion-cognition traits, physiological traits), and within-person fluctuations (amount of practice) jointly influence transfer. Across six rounds of a gamified learning task, participants ($N = 241$) trained under stress or control conditions and in consistent or inconsistent task environments. They then either continued or switched to the other task environment. Results revealed that task consistency enhanced efficiency during learning, but switching to an inconsistent environment disrupted performance. Patterns in pre- to post-switch performance were shaped by physiological reactivity and emotion-cognition traits, including cognitive reappraisal and intolerance of uncertainty, revealing compensatory adaptations that group-level analyses may obscure. These findings advance existing transfer models by highlighting how emotional and physiological regulation interact with environment.

The ability to transfer skills acquired in one context to a new context is a critical aspect of human learning[1]. Whether learning a new language or mastering a musical instrument, our capacity to apply learned skills in novel situations can define success. Despite the theoretical and practical importance of understanding the circumstances under which learning transfer occurs, there is little agreement among researchers on when skills generalize outside the learned context[1–3].

One reason for the lack of agreement is that researchers are inconsistent in how context is operationalized when examining near and far transfer[1]. Across domains, context denotes different things. In episodic memory, context often means the physical environment (e.g., underwater vs. on land; context-dependent memory)[4–6]. In motor and perceptual skill learning, context typically refers to sensorimotor/task constraints and training settings (e.g., simulation-based surgical skills[7], sport perceptual training[8]). In educational/cognitive training, context frequently refers to the similarity of stimuli or underlying cognitive operations (near vs. far transfer)[9–16]. In cognitive control/rule learning, context can denote stimulus–response or rule mappings that vary over time (variably mapped/dynamic tasks)[17,18]. Across these literatures, "transfer" refers to the extent to which performance and efficiency acquired in a trained context are maintained or appropriately adapted when contextual features change. More recently, a computational and neural framework has been proposed to unify generalization across such contexts[19], alongside classic individual-difference perspectives on skill learning[20].

Another reason for the lack of agreement is that researchers vary in which predictors they investigate. For example, psychologists may examine the roles of cognitive abilities[21–23], training[7,24], or sometimes both[25] on learning and skill transfer. While individual cognitive traits and training are well-documented predictors, emerging evidence suggests a broader set of determinants, including differences in training interventions, stress, personality, and fluctuations in emotional and physiological states[3]. These factors remain understudied but have critical implications for understanding the limits and potential of learning transfer.

Differing from the longstanding focus on training and cognitive abilities in the learning transfer literature, a recent review[3] suggests that accounts focusing solely on training quality or on learner traits are incomplete. When models omit interactions between task design, learner characteristics, and experiential histories, they risk understating transfer in subgroups where conditions and traits align. They can also obscure meaningful outcomes when positive and negative conditional effects cancel each other at the group

[1]Department of Psychological Sciences, Case Western Reserve University, Cleveland, OH, USA. [2]Booth School of Business, University of Chicago, Chicago, IL, USA. [3]Department of Psychological Sciences and Counseling, Youngstown State University, Youngstown, OH, USA. [4]School of Psychology, Georgia Institute of Technology, Atlanta, GA, USA. [5]Human Resources Research Organization (HumRRO), Alexandria, VA, USA. [6]Department of Psychological Sciences, Purdue University, West Lafayette, IN, USA. ✉e-mail: kyle.lafollette@chicagobooth.edu

level. An interactionist perspective on transfer emphasizes that the impact of a training intervention depends jointly on who is being trained, how, and for what future context, and highlights the value of modeling these cross-level contingencies explicitly[26]. Consistent with this view, we frame transfer as the outcome of a dynamic interplay among intervention-level factors, between-person differences, and within-person fluctuations.

At the **intervention level**, differences in the training task and conditions characterizing learning can significantly impact transferability[3]. Training programs that incorporate diverse and unpredictable challenges are better at preparing learners for novel situations than those relying on repetitive, highly structured practice[3,19,27–29]. This underscores the need for training designs that scaffold flexible, adaptive learning to bridge the gap between task-specific knowledge and real-world application[16,19,30,31].

**Between-person differences**, such as cognitive abilities, personality traits, and emotional-cognition traits also predict learning and performance and interact with other factors. Cognitive abilities play a well-documented role in shaping how well individuals learn with training[20,32,33]. Personality factors such as openness to experience, sensitivity to positive outcomes, and tolerance for ambiguity can influence learners' ability to generalize learned skills[26,34–37].

Traits such as emotional regulation, anxiety sensitivity, and stress tolerance influence decision-making, attentional control, and flexibility[38–42], all of which are crucial for successful learning and transfer. Individuals with high anxiety may avoid uncertain outcomes, leading to greater difficulty in learning and narrower transfer effects in novel situations[43,44]. For example, in a probabilistic learning task, highly anxious individuals may exhibit heightened threat anticipation and overestimate the likelihood of negative outcomes, leading them to rely on rigid, well-learned strategies rather than adapting flexibly to new contingencies[43]. Understanding how these emotion-cognition interactions drive variability in transfer outcomes could inform the design of more effective training interventions. By integrating these affective dimensions with traditional cognitive frameworks, researchers can better predict when and under what conditions skill transfer is most likely to occur.

The broad variability in individual learning and transfer trajectories, combined with high-stakes environments (e.g., surgery, battlefields), underscore the importance of identifying which individuals will show optimal transfer in novel situations. Generalizing from group-level analyses without considering between-person differences can obscure meaningful patterns in transfer that vary with trait-level differences[45]. This issue is particularly critical in skill transfer research, where predicting performance in new contexts depends on accounting for pre-transfer differences in skills and learning processes, highlighting the need to model individual trajectories accurately[45,46].

Furthermore, **within-person fluctuations**, such as variations in practice, affect, and physiological states with time may shape how individuals learn and transfer skills, and interact with intervention- and/or between-person-level factors. Some researchers have hypothesized that under conditions of certainty (e.g., when the correct response to a stimulus is consistent), individual differences tend to diminish with practice as learners converge on optimal strategies[20,47]. In contrast, under uncertain or variable conditions (e.g., when the correct response is less predictable), accumulating practice may amplify individual differences, as learners diverge in how they interpret and adapt to changing contingencies[20,47]. However, other studies have found evidence that variability across individuals can increase even under stable conditions[48]. These mixed findings underscore the need to examine learning and transfer not only in terms of average effects but also through the lens of individual-level trajectories and variability.

A complementary tradition in decision-making under uncertainty distinguishes between model-based (i.e., planning using an internal model or "cognitive map") and model-free (i.e., habitual, cached-value) control, with critical implications for transfer. Model-based control supports flexible generalization when contingencies change, whereas model-free control yields efficient performance in stable contexts but poorer transfer under revaluation or structural shifts (e.g., goal or transition changes)[49–51].

A successor representation stores a predictive map of which situations are likely to come next; it enables partial transfer when goals (rewards) change, but the underlying transition structure is preserved[52,53]. People shift between planning and habit when they judge that the benefits of being flexible outweigh the mental effort, and when the task makes planning feasible[54,55].

This literature is far richer on intervention-level manipulations than on trait moderators. Across perceptual, motor, and cognitive training, training variability robustly slows initial learning yet improves generalization, establishing task design as a primary driver of transferable learning[56]. In contrast, we know less about how stable traits and physiological regulation influence this flexibility. Stress, for example, can down-weight model-based control, with working memory buffering the effect[57]. Taken together, these perspectives predict a pattern of results: Training in consistent contexts promotes efficient routines, whereas switching into inconsistent contexts taxes planning/structure learning and reveals individual differences in flexibility.

In the present study, we investigated intervention-level manipulations, between-person traits, and within-person fluctuations as predictors of learning and learning transfer in a sample of undergraduate students and members of the local community. At the intervention-level, we manipulated task predictability (whether stimulus-response mappings were consistent vs. inconsistent) during training, change in task predictability following training, and acute stressors throughout. To examine individual differences, we measured emotion-cognition traits and baseline physiological stress responses. At the within-person level, we examined learning and learning transfer as a function of amount of practice, as well as changes in physiological markers of stress. Finally, we examined interactions among these variables. Methodologically, this study contributes to the transfer literature by combining objective performance and efficiency measures with physiological indices of stress reactivity, providing a process-oriented view of transfer that goes beyond self-report or coarse achievement outcomes.

We preregistered several hypotheses, separately for anticipated effects of learning during training (https://osf.io/wj68f) and transfer post-training (https://osf.io/x72z3). At the intervention level, we hypothesized that performance would be greater in the consistent condition than the inconsistent condition (H1), and that performance would be greater in a less stressful context than a more stressful context (H2). We further hypothesized that changes in amount of practice would interact with training condition (consistent vs. inconsistent) and post-training condition change (e.g., switching from consistent to inconsistent) to predict learning and transfer (H3). Finally, we hypothesized that several emotion-cognition traits at the between-person (i.e., individual differences) level would interact with training condition and stress context at the intervention level, including elements of emotion regulation, intolerance of uncertainty, and behavioral inhibition (H4). Through a series of models, we demonstrate the contributions of each level of analysis (i.e., intervention, between-person, within-person) in understanding individual differences in acquisition and transfer of learned skills, and how those differences deviate from group-level findings.

This study seeks to advance transfer theory by jointly manipulating task structure and stress while modeling between-person traits and within-person practice. Consistent contexts should foster efficient, model-free routines that transfer poorly to volatile settings; inconsistent contexts should recruit model-based planning, supporting transfer when contingencies change but taxing learners unevenly depending on stress reactivity and emotion-cognition traits. For practitioners, this multi-level account yields potential opportunities for training variability (e.g., switching between consistent and inconsistent environments), stress context intervention (e.g., practicing under stress), and trait-informed scaffolds (e.g., identifying favorable training environments based on individual traits and preferences).

## Methods
### Participants
Two-hundred sixty-five adults were recruited for the experiment. Nine chose not to participate, leaving two-hundred fifty-six adult participants.

Half the participants were recruited from the university's SONA subject pool, the other half from the local Cleveland community, in exchange for course credit or $80 USD, respectively. All participants were required to be between 18 and 30 years of age, proficient in the English language, and have no self-reported history of heart disease, peripheral vascular disease, venous thrombosis, diabetes mellitus, fainting/loss of consciousness, seizures, frostbite, loss of feeling in limbs, or Reynaud's syndrome. Cardiology researchers have reported that measures such as heart-rate variability change with age[58,59], so we restricted the sample to 18–30 years of age to minimize age-related heterogeneity in autonomic physiology that could confound stress-reactivity and transfer estimates in our sample. In our preregistration, we set a target of $N = 252$ based on three considerations: prior work using the same task showed significant effects with $N = 128$; we planned individual-difference analyses and aimed for approximately 60 participants per between-subjects condition to support those associations; and we anticipated modest data loss and therefore buffered the target to maintain planned analytic power.

Participants were eligible to take part in the experiment only if they had normal color vision and were not taking a medication that decreased cold sensitivity or influenced heart rate variability (HRV). The study employed a $2 \times 2 \times 2$ between-subjects design, manipulating Training Version (consistent, inconsistent), Stress Condition (stress, control), and Post-Training Version (same, switch). Fifteen participants were excluded from data analysis due to missing data (e.g., missing the final round due to slow performance, poor readouts from the ECG data), for a final sample of 241 (143 women, 94 men, 4 non-binary, mean age = 22.44, SD = 3.97). The sample was comprised of 4 participants identifying as American Indian, First Nations, or Alaskan native, 103 participants identifying as Asian/Asian American, 29 participants identifying as Black, African American, or African Caribbean, 1 participant identifying as Native Hawaiian or other Pacific Islander, 104 participants identifying as White/European or Middle Eastern decent, 22 participants identifying as Hispanic/Latinx, and 4 participants who chose not to respond (participants could report more than one racial and/or ethnic identity). This experiment received approval from Case Western Reserve University's Institutional Review Board (STUDY20220308).

## Material
### Questionnaires
All participants completed a demographic questionnaire including self-reported gender, race, ethnicity (no analyses were preregistered or conducted for these variables), and four computerized questionnaires administered via REDCap (Research Electronic Data Capture[60]), designed to assess key emotion-cognition traits. These measures were chosen because they capture fundamental psychological processes that integrate emotional and cognitive functioning, providing valuable insights into individual differences in emotion regulation, sensitivity to uncertainty, anxiety, and motivational systems. Questionnaire responses demonstrated moderate to strong internal reliability (α, ω) and small to moderate intercorrelations with conceptually related scales (e.g., IUS-12 and BIS). See Table 1.

The measures used included the Intolerance of Uncertainty Scale—Short Form (IUS-12[61]), which evaluated individuals' difficulty in tolerating uncertainty, a key cognitive-emotional trait associated with heightened anxiety, impaired decision-making, and rigid thinking patterns. The Behavioral Inhibition System and Behavioral Activation System Scales (BIS-BAS[62]) assessed two motivational systems: The Behavioral Inhibition System (BIS), reflecting sensitivity to punishment and avoidance behavior, and the Behavioral Activation System (BAS), measuring sensitivity to reward and approach behavior, providing insights into emotional and motivational responses. The Emotion Regulation Questionnaire (ERQ[63]) measured two primary strategies of emotion regulation: cognitive reappraisal, which involves modifying thoughts to change emotional impact, and expressive suppression, which involves inhibiting outward emotional expressions, offering insights into how individuals manage emotions for psychological well-being and cognitive functioning. The State-Trait Anxiety Inventory (STAI[64]) differentiated between state anxiety, a temporary emotional response to situational stress, and trait anxiety, a stable predisposition to experience anxiety, providing a comprehensive perspective on the influence of anxiety on cognitive and emotional processes. Last, a NASA-TLX (Task Load Index[65]) was administered to assess perceived workload at the conclusion of the experiment.

### Learning task
We administered a six-round learning task that required participants to manage structures on a grid-world, gather resources, and defend against invaders[17]. Such simulated environments can support skill transfer when they capture structural features of target tasks (e.g., variability, feedback, time-pressure). The learning task was created with and administered via E-Prime 3[66] on Dell computers at a resolution of 1920 × 1080.

The task involved controlling an avatar on a computer screen to complete two separate missions. The two missions differed in their goals (seeking rewards vs. avoiding losses) and optimal strategies (reactively pursuing energy sources vs. proactively managing threats), enhancing the generalizability of observed effects. In the resource collection mission, participants aimed to gather energy from suns moving across the screen by planting sunflowers to collect their energy (Fig. 1A). The suns varied in size and color, indicating their energy output and speed, respectively. In the defense mission, participants had to combat zombies to prevent them from infiltrating a town (Fig. 1B). Zombies also varied in size and clothing color, reflecting their toughness and speed, respectively, and could be defeated by planting pea shooters that fired at them. Participants needed to strategically plant sunflowers and pea shooters to maximize energy collection and defeat zombies. Participants completed the resource collection mission followed by the defense mission in each round. Participants completed five rounds.

Participants were randomly assigned to one of two task versions. The two versions of the task (consistent vs. inconsistent) were identical with the exception that the consistent version's stimulus-response mappings were maintained throughout the task (e.g., red suns were always fastest, large suns

## Table 1 | Mean and standard deviation of questionnaire scores, and intercorrelations between participant scores

| Variable | M | SD | α | ω | 1 | 2 | 3 | 4 | 5 | 6 | 7 |
|---|---|---|---|---|---|---|---|---|---|---|---|
| 1. IUS -12 Intolerance of Uncertainty | 29.93 | 7.60 | 0.85 | 0.86 | – | | | | | | |
| 2. BAS Drive | 11.15 | 1.98 | 0.74 | 0.78 | 0.13 | – | | | | | |
| 3. BAS Reward Responsiveness | 17.19 | 1.83 | 0.68 | 0.71 | **0.16*** | **0.45*** | – | | | | |
| 4. BAS Fun Seeking | 11.78 | 2.04 | 0.65 | 0.69 | −0.09 | **0.37*** | **0.37*** | – | | | |
| 5. BIS | 19.63 | 2.15 | 0.75 | 0.81 | **0.32*** | **016*** | **0.29*** | **−0.15*** | – | | |
| 6. ERQ Expressive Suppression | 15.63 | 5.02 | 0.74 | 0.77 | **0.19*** | −0.10 | −0.12 | −0.07 | −0.01 | – | |
| 7. ERQ Cognitive Reappraisal | 29.59 | 5.74 | 0.83 | 0.87 | **−0.19*** | **0.13*** | 0.16 | **0.14*** | -0.05 | −0.10 | – |
| 8. STAI Trait Anxiety | 22.68 | 9.95 | 0.90 | 0.91 | **0.45*** | −0.07 | 0.01 | 0.01 | **0.29*** | 0.09 | **−0.34*** |

Bolded text indicates a significant effect with inference criteria $p < 0.05$. * $p < 0.05$, ** $p < 0.01$, *** $p < 0.001$, α = Cronbach's alpha, ω = McDonalds' total omega.

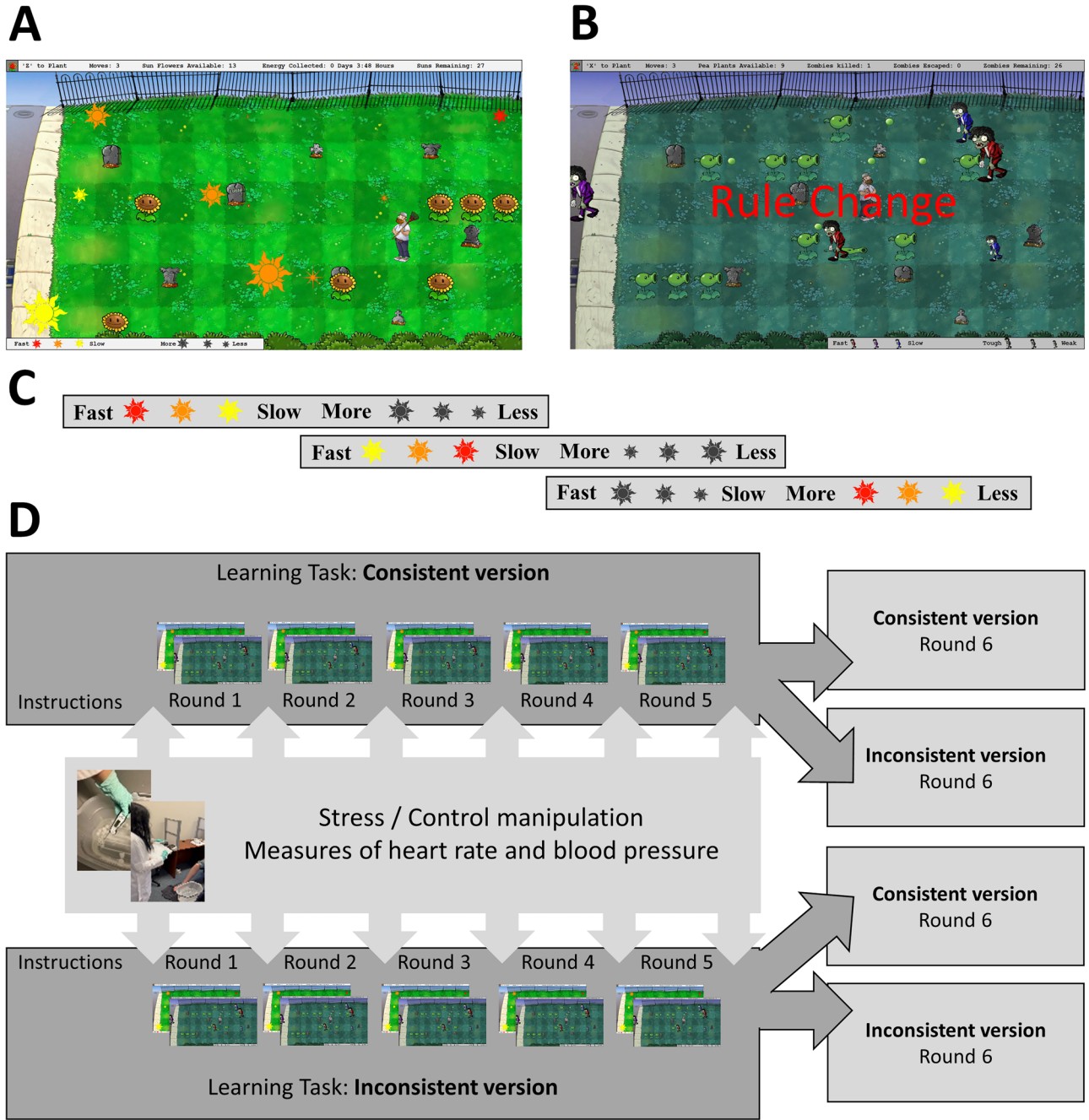

**Fig. 1 | Overview of the experimental task and design. A**, **B** illustrate the two task missions: **A** the resource collection mission, where participants collect energy by planting sunflowers in the path of moving suns, and **B** the defense mission, where participants defend a town by strategically planting pea shooters to combat zombies. **B** also depicts the inconsistent version of the task's rule change, occurring every 50 moves. **C** depicts example variations of these rule changes in the resource collection mission, manipulating stimulus speed and value. **D** shows the sequence of stress and control manipulations applied across six rounds of the learning task. In the first five rounds, participants either complete the consistent version or inconsistent version. In the sixth round, half of the participants switch to the alternate task version to assess learning transfer. Behavioral and physiological measures (heart rate, blood pressure) were recorded throughout. Participants in the stress condition completed a variant of the Maastricht Acute Stress Test (MAST) prior to each round, while those in the control group completed a neutral version of the test.

always produced the most energy), whereas the inconsistent versions' stimuli-response mapping were variable (e.g., red might indicate the fastest sun, then after 50 moves red might indicate a sun that produces the least amount of energy, then the slowest, and so on; see Fig. 1C). Stimuli-response mappings were always presented at the bottom of the screen in both versions such that there was not an additional memory load in the inconsistent version. Likewise, the stimuli would visibly change to match the correct mapping such that identical strategies across the two versions would yield identical scores. This manipulation was designed to examine how unpredictability influences learning, adaptability, and performance.

A critical aspect of this experiment involved a surprise sixth and final round, where half of the participants were switched to the version of the task on which they were not trained (e.g., consistent to inconsistent), while the other half continued with their trained version (e.g., consistent to consistent). This manipulation was central to investigating learning transfer, specifically whether participants' scores or completion times would exhibit partial or complete transfer upon switching as a function of their trained version. Performance was measured by the total energy collected or points earned from defeating zombies (score) and the time elapsed during the round (efficiency).

Performance for each mission was z-scored, using as the reference the Round 1 performance of participants in the Control, Consistent group (i.e., the theoretically easiest condition). The z-scores were averaged together to get one standardized performance score per round. Completion time was z-scored using Round 1 completion time across missions of participants in the Control, Consistent group as a reference for analyses. Raw time in minutes are reported in figures. Additional details about this task can be found in Macnamara and Frank[17].

## Procedure

After obtaining informed consent, each participant removed their socks and shoes and wore rubber sandals. Participants were fitted with disposable snap electrodes (Biopac EL507) connected via leads to a Biopac BioNomadix 2-channel wireless amplifier and accompanying AcqKnowledge software to measure electrocardiogram (ECG) signals. ECG was collected continuously through the experiment and sampled at 2000 Hz from three electrodes: one attached to the upper right clavicle just below the collarbone, and another two attached just below the lowest left and right ribs. The electrode attached to the right rib was used as a reference. Participants also wore a breathing belt over their clothing, positioned around the torso under the armpits.

After affixing the electrodes and respiration belt, participants completed the questionnaires in the following order: demographics, IUS-12, BIS-BAS, ERQ, STAI. Next, participants were instructed to fix their gaze on a centrally located cross for 5 min. This procedure was used to ensure accurate readings and establish baseline measures for heart rate and respiration. Following this, their blood pressure was measured using a standard non-prescription blood pressure cuff. These baseline measures were used as references for calculating standardized changes from baseline in all HRV and blood pressure analyses.

Participants then read instructions for their version of the learning task. Next, half the participants in each task version underwent a stress manipulation before each round, while the other half experienced a control manipulation (Fig. 1D). Those in the stressful environment completed a variant of the Maastricht Acute Stress Task (MAST[67]) which combined a cold pressor, serial subtraction, and social-evaluative threat by an experimenter wearing a white lab coat with flat affect. Participants were instructed to submerge one foot in ice water maintained at approximately 2 degrees Celsius for an undisclosed duration (between 45 and 75 s), fixed across participants. After the cold pressor, participants were given a four-digit number and asked to subtract a two-digit prime number from it for 60 s. They then re-submerged their foot in ice water for another variable period. Conversely, participants in the control condition used lukewarm water at approximately 36 degrees Celsius and were asked to simply count upwards from zero by a friendly experimenter in a blue lab coat.

Participants experienced a round of the MAST (or control version of the MAST) before completing one round of the learning task (energy collection mission followed by the enemy defense mission). Participants completed the MAST and learning task round procedure five times and then completed the surprise sixth round either in the same version or switched versions. Finally, participants completed a final resting period and the NASA-TLX.

## Electrocardiogram processing and analysis

Cold-pressor–based stressors, including the MAST, robustly engage both the autonomic nervous system and the hypothalamic–pituitary–adrenal (HPA) axis, with numerous studies reporting elevations in salivary cortisol following these tasks[68–71]. In the present study, we measured autonomic indices of stress reactivity: heart rate and mean arterial pressure (primarily reflecting sympathetic activation) and heart-rate variability metrics: Root Mean Square of Successive Differences (RMSSD) and high-frequency power as markers of parasympathetic/vagal activity, complemented by cardiac sympathetic index (CSI) and low-frequency power as sympathetic proxies and cardiovagal index (CVI) as a parasympathetic proxy[72–74].

ECG data were processed and analyzed using 5-min segments from the baseline period and during each of the six MAST sessions, resulting in seven HRV features per participant and window. Features were extracted using the Python library Neurokit2[75]. ECG signals are susceptible to contamination from multiple sources, including interference electromagnetic interference from power lines, channel disruptions, muscle activity noise, issues with electrode contact, and baseline shifts resulting from body movements or breathing patterns. To address these challenges and improve the accuracy of feature extraction, the ECG signals were cleaned first using Neurokit2.

We first detrended the signal using a 0.5 Hz highpass 5th order Butterworth filter, effectively suppressing low-frequency components below 0.5 Hz and eliminating baseline wander. We then denoised the signal by filtering out 60 Hz powerline noise, smoothing it with a moving average kernel with a width of one period of 60 Hz. From these processed signal data, we identified R-peaks using a gradient-based approach[75]. First, we computed the gradient of the ECG signal, which helps highlight rapid changes characteristic of QRS complexes. To reduce noise and improve detection stability, we then applied smoothing using a boxcar filter over two windows: a short-term window of 200 ms to reduce high-frequency fluctuations, and a longer-term window of 1.5 s to compute a stable average gradient. A dynamic threshold for identifying QRS complexes was then set by scaling this smoothed gradient with an adjustable weight, set to 1.5 times the smoothed gradient in our analyses. We detected the beginning and end of QRS complexes based on whether the smoothed gradient exceeded the threshold, discarding short detections. Within each retained QRS segment, we identified local maxima in the original ECG signal and selected the most prominent peak as the R-wave. RR-intervals of the excluded beats were corrected using linear interpolation, and physiological outliers were identified if the heart rate dropped below 45 bpm or exceeded 200 bpm. As a final effort to ensure data quality, we subjected the identified R-peaks to an artifact removal algorithm developed by Lipponen and colleagues[76] to further detect outliers on the basis of R-R intervals.

Cleaned R-peak data, represented as an array of ECG intervals, were used to compute HRV features, including average heart rate and the RMSSD, both of which are derived from normal-to-normal intervals—representing the time between successive R peaks. We also conducted spectral analysis on the interpolated R-R intervals using Welch's method to calculate the high frequency (0.15–0.40 Hz; HF) and low frequency (0.04–0.15 Hz; LF) components of HRV. These values reflect the power in each band, indicating autonomic nervous system dynamics. Additionally, two indices were calculated as potentially more accurate proxies for autonomic nervous system activity: the CSI and the CVI. These indices were derived from Lorenz plots, which transform R-peak variability into an elliptical distribution. The longitudinal (L) and transverse (T) axes of this ellipse were measured, and the indices were calculated using the formulas $CVI = \log_{10}(L \times T)$ and $CSI = L/T$. Previous research has found that CVI is sensitive to parasympathetic blockade by atropine, whereas CSI is sensitive to sympathetic blockade by propranolol[73].

## Data transparency statement

All hypotheses, methods, and analyses for the Rounds 1–5 models were preregistered at https://osf.io/wj68f on January 5th, 2023, prior to data collection. Hypotheses, methods, and analyses for the Rounds 5 and 6 transfer models were preregistered at https://osf.io/x72z3 on January 5th, 2023, again, prior to data collection. Next, we describe deviations from our preregistrations:

1. Expanded Individual Difference Measures. **Deviation**: Subscales of emotion-cognition traits beyond the overall measure named in the preregistration (e.g., cognitive reappraisal, fun-seeking) were included in the analyses. **Justification**: These additional, more targeted measures, collected to complement the preregistered traits (e.g., BAS, anxiety, intolerance of uncertainty), provided critical information for understanding individual variability. Their inclusion allowed for more nuanced interpretations of performance and efficiency across task conditions and stress manipulations. These subscale scores were treated in the same manner as their preregistered total scores (i.e., same criteria for outlier detection and correction, same incorporation in linear mixed effects models, etc.).

2. Nonlinear and Physiological Modeling. **Deviation**: Preregistered hypotheses emphasized linear relationships between stress, traits, and performance. However, we incorporated curvilinear trends (e.g., round-squared terms) and expanded physiological analyses, including change in heart rate, CSI, and vagal tone (CVI), to better capture stress responses. Though we included curvilinear trends in the preregistration, we did not specify curvilinear effects. **Justification**: Curvilinear trends (e.g., performance leveling off over time) and supplementary physiological indices provided greater nuance in characterizing individual differences in learning and stress reactivity. These adjustments provided a more comprehensive understanding of the interplay between stress, physiology, and performance.

3. Post Hoc Analyses of Emotion Regulation, Intolerance of Uncertainty, and BIS-BAS. **Deviation**: Exploratory post hoc transfer analyses examined emotion-cognition traits and their interaction with performance, efficiency, and switching beyond the scope of preregistered hypotheses. **Justification**: While these individual difference measures were included in the preregistration, we did not consistently specify directional hypotheses about their effects. However, consistent patterns emerged during the transfer phase that warranted further exploration. These analyses helped clarify and contextualize the observed results.

4. NASA-TLX analyses not conducted. **Deviation**: The NASA-TLX was administered at the conclusion of the experiment and analyses preregistered to determine effects of individual differences traits and stress on mental effort/workload. **Justification**: These analyses are outside the scope of the present study report.

## Results

The primary goals of this experiment were to investigate the roles of intervention, between-person, and within-person level factors on learning and transfer to new contexts. The variables of interest in this experiment include round (to capture linear change), round squared (to capture curvilinear change), task version (consistent, inconsistent), switch condition (switched to the untrained version or remained in the trained version), stress condition (Maastricht Acute Stress Test [MAST], control version of the MAST), and change in heart rate, HRV, and blood pressure from baseline/rest to each MAST administration. We also examined baseline HRV as a trait-level marker of stress response.

Unless noted otherwise, all hypotheses were tested with linear-mixed effects models using the *statsmodels*[77] Python library. All tests were two-sided. Round and round squared were entered as within-subject fixed effects; task Version (consistent, inconsistent), and Stress condition (MAST, control) were entered as between-subject fixed effects. All two- and three-way interactions were included except interactions that simultaneously contained both round and round squared. Participant-specific random intercepts captured between-person differences in baseline level. Physiological predictors were operationalized as standardized change from baseline at each administration immediately preceding the round; performance scores were standardized and completion time was measured in minutes. We report coefficients with 95% confidence intervals and *p* values.

### Manipulation checks

As a manipulation check for our MAST procedure, we examined the effects of stress condition on physiological measures while controlling for other task-related variables. We conducted three separate linear mixed effects models predicting standardized change from baseline in (a) HRV indexed by the RMSSD, (b) heart rate, and (c) mean arterial blood pressure (MAP). Round and round squared were included as within-subjects fixed effects, and stress condition and task version were included as between-subjects fixed effects. All two- and three-way interactions were included as fixed effects except for any interactions that would include both round and round squared.

We found no significant change in HRV as a function of stress condition ($\beta = -0.098$; 95% CI [−0.295, 0.098]; $p = 0.326$). The lack of a manipulation effect on HRV may suggest that variability was not sensitive to our manipulation or is possibly more informative as a marker of individual

adaptive capacity[72,78,79]. However, we did observe a significant positive effect of stress condition on standardized change in heart rate from baseline ($\beta = 0.360$, 95% CI [0.125, 0.594], $p = 0.003$) and a significant positive effect of stress condition on standardized change in MAP from baseline ($\beta = 0.554$, 95% CI [0.350, 0.758], $p < 0.001$). This indicates that participants experienced a strong increase in vasoconstriction in response to the stress manipulation.

To better understand whether our manipulation targeted the sympathetic or parasympathetic nervous systems, we calculated supplementary frequency-based and non-linear time-based measures of ECG activity. These included two measures of sympathetic tone, standardized change in low frequency power (LF) and CSI, and two measures of parasympathetic tone, standardized change in high frequency power (HF) and cardiac vagal index (CVI). The high frequency band of ECG is a well-documented marker of parasympathetic activity and vagal tone; however, researchers disagree as to the interpretation of the low frequency band[74]. CSI and CVI, however, have been established as sensitive to sympathetic and parasympathetic blocking agents, respectively. Administration of propranolol, a sympathetic nervous system beta blocker, tends to correspond with a decrease in CSI, whereas administration of atropine, a parasympathetic blocker, tends to decrease CVI[73]. We find significant positive effects of our stress manipulation on both change in low frequency power and CSI, but not high frequency power and CVI (see Table 2). We also find that high frequency spectral power decreases over rounds in the control condition, but not in the stress condition. Together, these results suggest that our manipulation targeted sympathetic tone without an effect on the parasympathetic nervous system (though parasympathetic response indexed by HF did change across rounds, with participants in the control condition having an initial HF response that quickly subsided, whereas participants in the stress condition had a more sustained HF response across rounds).

### Performance and efficiency

Following our demonstrated effect of the stress manipulation, we next aimed to explore how both stress and task characteristics together influenced task performance during the skill acquisition phase of the task (Rounds 1–5). We conducted two separate linear mixed effects models predicting performance scores and efficiency over the first five rounds of the learning task. Round and round squared were included as within-subjects fixed effects, and stress condition and task version were included as between-subjects fixed effects. Round and round squared terms were tested to investigate linear and curvilinear growth. However, models investigating exponential and power trajectories of the form $e^{-\lambda x}$ and $x^{-\lambda}$ are reported in SI with near identical results. Also see SI for models including mission type as a standalone fixed effect, in which we demonstrate that, as predicted, mission has no statistically significant effect on performance nor completion time, resulting in its exclusion from models reported here. We preregistered primary models and report them in order below. Table 3 maps each research question to the corresponding model, predictors, interactions, and outcomes.

We observed main effects of round ($\beta = 0.348$, 95% CI [0.221, 0.476], $p < 0.001$; Fig. 2A) and round squared ($\beta = -0.043$, 95% CI [−0.073, −0.012], $p = 0.006$) on performance scores, demonstrating linear and curvilinear growth reflective of typical learning curves (see Fig. 2). Similarly, we observed main effects of round ($\beta = -0.672$, 95% CI [−0.774, −0.570], $p < 0.001$) and round squared ($\beta = 0.107$, 95% CI [0.082, 0.131], $p < 0.001$) on completion time. Neither stress condition nor task version had any significant effect on performance scores nor completion time (no support for H1 and H2 during acquisition). Overall, our models explained 39.02% of the variance in performance and 48.76% of the variance in completion time.

We next tested the effects of switching task versions at Round 6 on both performance scores and completion times. Two separate linear mixed effects models were conducted over the final two rounds of the task. These models included all predictors from previous analyses (excluding round squared as these models included only two time points), with the addition of the factor Switching (whether participants switched task versions between

**Table 2 | Linear mixed effects model results predicting changes from baseline in physiological indices of stress reactivity**

| Predictor | Primary Measures | | | Parasympathetic | | Sympathetic | |
|---|---|---|---|---|---|---|---|
| | Change in RMSSD | Change in MAP | Change in HR | Change in CVI | Change in HF | Change in CSI | Change in LF |
| (Intercept) | 0.054 (−0.084, 0.192) | **−0.248 (−0.391, −0.105)**\** | **−0.179 (−0.344, −0.014)**\* | 0.014 (−0.135, 0.164) | −0.052 (−0.220, 0.116) | −0.113 (−0.261, 0.034) | −0.113 (−0.272, 0.047) |
| Stress Condition | −0.098 (−0.295, 0.098) | **0.554 (0.350, 0.758)**\*** | **0.360 (0.125, 0.594)**\** | −0.025 (−0.237, 0.188) | 0.098 (−0.140, 0.336) | **0.240 (0.029, 0.450)**\* | **0.231 (0.004, 0.457)**\* |
| Round | 0.137 (−0.054, 0.328) | −0.062 (−0.229, 0.106) | **−0.185 (−0.293, −0.078)**\** | 0.151 (−0.012, 0.313) | **−0.182 (−0.297, −0.076)**\** | 0.155 (−0.008, 0.317) | 0.012 (−0.112, 0.137) |
| Round Squared | −0.105 (−0.296, 0.086) | **0.194 (0.028, 0.361)**\* | 0.026 (−0.081, 0.134) | −0.050 (−0.212, 0.113) | **0.130 (0.024, 0.235)**\* | −0.116 (−0.279, 0.046) | −0.015 (−0.140, 0.110) |
| Stress Condition * Round | −0.052 (−0.326, 0.222) | 0.057 (−0.184, 0.297) | **−0.257 (−0.411, −0.078)**\** | −0.005 (−0.238, 0.228) | **0.242 (0.091, 0.394)**\** | **−0.269 (−0.502, −0.036)**\* | −0.026 (−0.205, 0.153) |
| Stress Condition * Round Squared | 0.144 (−0.130, 0.418) | −0.086 (−0.327, 0.155) | 0.113 (−0.041, 0.268) | 0.062 (−0.172, 0.295) | **−0.173 (−0.324, −0.021)**\* | 0.125 (−0.108, 0.358) | −0.034 (−0.213, 0.145) |

Standardized weights are reported alongside 95% confidence intervals in parentheses. Bolded text indicates a significant effect with inference criteria $p < 0.05$. * indicates $p < 0.05$. ** indicates $p < 0.01$. *** indicates $p < 0.001$. RMSSD root mean square of successive differences, MAP mean arterial blood pressure, HR average heart rate, CVI cardiac vagal index, HF high frequency power. CSI cardiac sympathetic index, LF low frequency power.

**Table 3 | Analysis map of hypotheses (H1–H4) to models, predictors, interactions, and outcomes**

| Hypothesis | Phase/Dataset | Model | Key Predictors (fixed) | Key interactions | Outcome(s) | Prereg. |
|---|---|---|---|---|---|---|
| H1: Consistent > Inconsistent during acquisition (performance, efficiency) | Acquisition (Rounds 1–5) | LME: y ~ Round + Round² + Version + Stress + (1\|id) | Round, Round²; Version (Certain vs Uncertain); Stress (MAST vs Control) | Round × Version; Round × Stress; Version × Stress | Standardized Performance; Completion Time (min) | Yes |
| H2: Lower stress > Higher stress (performance, efficiency) | Acquisition (Rounds 1–5) | Same LME as H1 (reported jointly) | As above | As above | Standardized Performance; Completion Time | Yes |
| H3: Practice interacts with training & post-training change to predict transfer | Transfer (Round 6 vs Round 5) | LME: y ~ Phase (R5/R6) + Training Version + Switch (Yes/No) + Phase×Version + Phase×Switch + Version×Switch + (1\|id) | Phase (pre/post); Version; Switch | Phase × Version; Phase × Switch; Version × Switch; Phase × Version × Switch | ΔPerformance (R6 − R5); ΔTime (R6 − R5) or Level at R6 with R5 covariate | Yes |
| H4: Emotion–cognition traits moderate acquisition & transfer | Acquisition (R1–R6) and Transfer (R5 → R6) | LME (Acq): y ~ Round + Round² + Version + Stress + Trait + 2-way/3-way interactions + (1\|id)LME (Trans): Δy ~ Version + Switch + Trait + interactions + (1\|id) | Trait (e.g., ERQ reappraisal, IUS-12, BIS/BAS); Version; Stress; Round/Phase | Trait × Version; Trait × Stress; Trait × Round (or Trait × Phase); selected 3-ways (e.g., Trait × Version × Phase) | Standardized Performance; Completion Time | Yes |

LME Linear Mixed-Effects model, Version training environment (Consistent vs Inconsistent), Stress MAST vs Control, Trait emotion–cognition trait, ERQ reappraisal emotion regulation questionnaire: reappraisal, IUS-12 Intolerance of Uncertainty, BIS/BAS behavioral inhibition system/behavioral activation system scales, Phase pre/post switch (Round 5 vs 6).

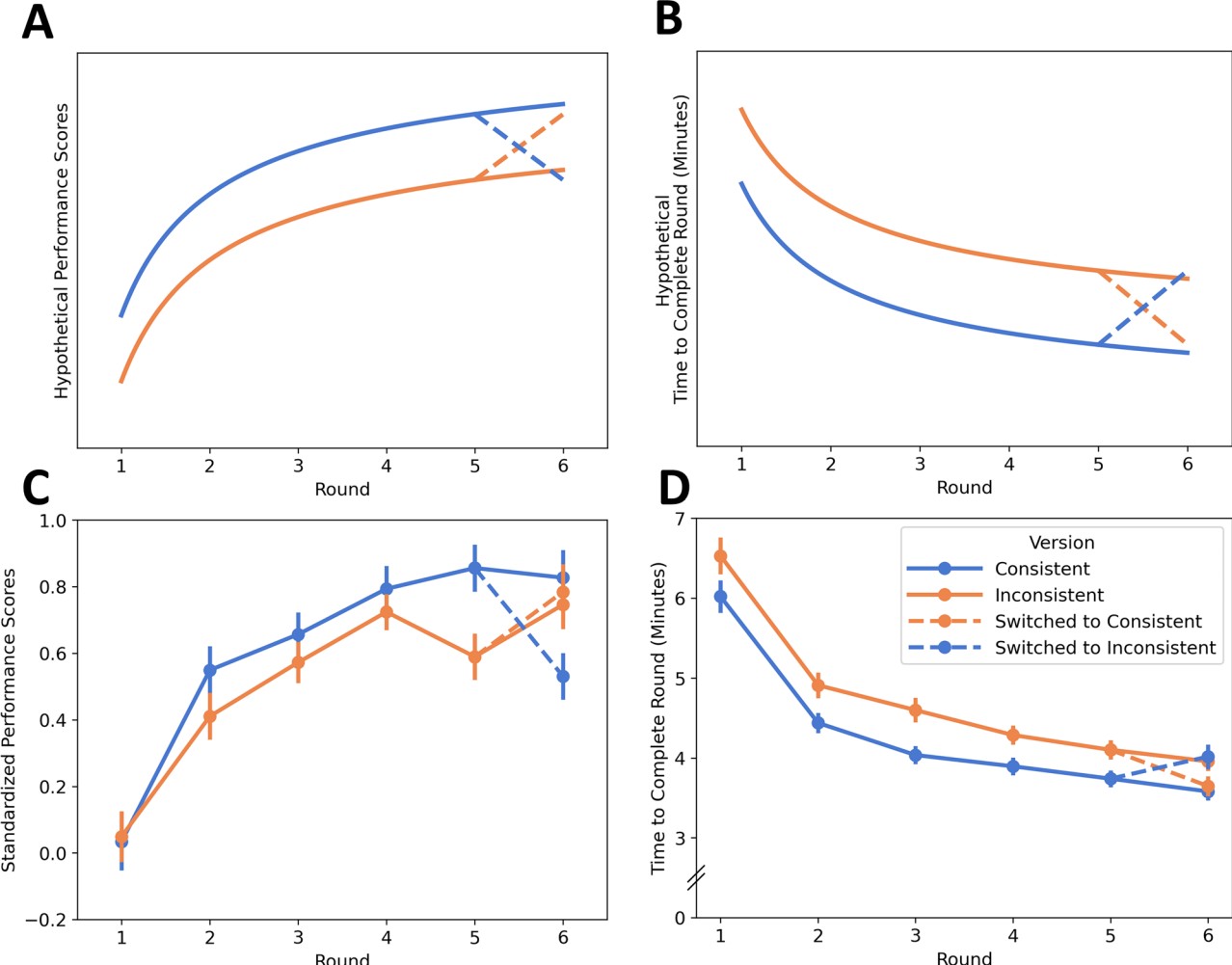

**Fig. 2 | Standardized performance scores and task completion times across six rounds under different training conditions, including transfer effects in Round 6.** The top row (**A**, **B**) shows hypothesized trends, while the bottom row (**C**, **D**) presents empirical data. Hypothetical trends (**A**) show increasing performance scores plateauing by Round 5, with a decline in Round 6 after switching from the consistent to the inconsistent version and an increase after switching from the inconsistent version to the consistent version. Hypothetical task completion times (**B**) show steady decreases with practice, with times converging to the untrained version's pace after switching in Round 6. In the empirical data, performance scores

(**C**) increase across rounds, plateauing by Round 5 (Round: $\beta = 0.348$, $p < 0.001$; Round Squared: $\beta = -0.043$, $p = 0.006$), with a decline in Round 6 after switching from the consistent to the inconsistent version and an increase after switching from the inconsistent version to the consistent version. In the empirical data, task completion times (**D**) consistently decrease (Round: $\beta = -0.672$, $p < 0.001$; Round Squared: $\beta = 0.107$, $p < 0.001$), with complete transfer evident in Round 6 as times match those of the untrained version. Error bars represent $\pm 1$ SE. $N = 241$ participants.

Rounds 5 and 6) and its interactions with stress condition, task version, and round. Nine participants were excluded due to missing data on Rounds 5 and 6, leaving $n = 232$ for analysis. We observed a significant negative Switching × Round interaction on performance score ($\beta = -0.386$, 95% CI [−0.665, −0.108], $p = 0.004$), indicating that participants who switched versions tended to perform worse relative to those who remained in the same task version—particularly among those switching from the consistent to the inconsistent version—suggesting a disruption in performance associated with the shift in context (Fig. 2C; supporting H4).

For completion time, we observed a significant positive Switching × Round interaction ($\beta = 0.242$, 95% CI [0.111, 0.372], $p < 0.001$) and a significant negative Switching × Round × Training Version interaction ($\beta = -0.373$, 95% CI [−0.568, −0.177], $p < 0.001$). These interactions indicate that participants who switched task versions adjusted their efficiency—speeding up or slowing down—so that by Round 6, their completion times closely resembled those of participants who had continued with the same version. Thus, while performance scores suggest incomplete or disrupted transfer, the convergence in completion times is consistent with efficient

behavioral adaptation, potentially reflecting partial or domain-specific transfer of task structure or motor routines (Fig. 2D; supporting H4).

### Individual differences in stress reactivity

We repeated these analyses with three model variations, each time replacing the stress condition variable with one of our three physiological indices of stress reactivity: standardized change in MAP from baseline, standardized change in heart rate from baseline, and standardized change in HRV from baseline. These change measures were collected during the MAST procedure immediately preceding each learning task round. Effects of round and round squared remained significant throughout these model variations. A small positive effect of task version on completion time emerged across the models, with $\beta$ ranging between 0.174 and 0.223 (see Supplementary Tables S1–S8 for full model outputs; partial evidence supporting H1). We also observed effects of stress specific to change in blood pressure and heart rate. For participants trained on the inconsistent version of the task, those who experienced a greater positive change in MAP relative to baseline initially completed the task more slowly (Training Version

(Inconsistent) × MAP Change: $\beta = 0.143$, 95% CI [0.030, 0.257], $p = 0.013$), but had greater acceleration over rounds (Training Version (Inconsistent) × MAP Change × Round: $\beta = -0.127$, 95% CI, $-0.240$ to $-0.013$; $p = 0.028$). Furthermore, participants who experienced a greater positive change in heart rate relative to baseline had a higher performance score if trained on the consistent version of the task (Heart Rate Change: $\beta = 0.141$, 95% CI [0.027, 0.255], $p = 0.016$) but not the inconsistent version (Training Version (Inconsistent) × Heart Rate Change: $\beta = -0.156$, 95% CI [$-0.310$, $-0.002$], $p = 0.046$).

Although we did not observe an effect of stress condition on HRV, nor did we observe any effects of stress condition on standardized performance scores or time to completion, there remained a possibility that individual differences in HRV might predict the magnitude of a stress effect. In other words, people with greater HRV might show greater resilience to stress[80,81], manifesting in better and faster performance relative to people with lower HRV. To test for this potential interaction at the transfer stage, we conducted two linear mixed effects models predicting change in performance and efficiency over the final two rounds of the learning task. The predictors in these models were identical to those in the previous transfer models, with the addition of change in RMSSD relative to baseline (as a measure of change in HRV) in place of stress condition.

In addition to the previously observed effects (e.g., Switching × Round and Switching × Round × Training Version) on performance score and efficiency, we observed multiple effects of RMSSD on performance and completion time. With respect to performance, participants who stayed with the inconsistent version of the task between Round 5 and 6 had a notably negative effect of change in RMSSD on performance (Switching × Round × Training Version × RMSSD Change: $\beta = 0.833$, 95% CI [0.080, 1.586], $p = 0.030$). With respect to completion time, although we observe a generally negative overall effect of RMSSD on completion time ($\beta = -0.264$, 95% CI [$-0.478$, $-0.049$], $p = 0.016$), participants who stayed with or switched to the consistent version of the task between Round 5 and 6 had a notably positive effect of change in RMSSD on completion time (Switching × Round × Training Version × RMSSD Change: $\beta = 0.402$, 95% CI [0.040, 0.764], $p = .029$). See Fig. 3. Taken together, these findings suggest that HRV, specifically, increases in RMSSD from baseline in Rounds 5 to 6, may index adaptive physiological regulation that facilitates behavioral adaptation and transfer under stress. Greater increases in RMSSD were associated with slower decision-making when transitioning into a more stable, consistent environment, and with poorer performance when participants faced continued task inconsistency. These results align with prior work linking high HRV to cognitive flexibility and emotion regulation capacity[80], and they suggest that physiological adaptability may shape how individuals respond to dynamic changes in task structure and environmental demands.

## Individual difference in emotion-cognition traits

Although the previous findings offer valuable insights into performance and efficiency in learning, as well as learning transfer, they may overlook the influence of individual differences in emotion-cognition traits. Without accounting for these traits, such as emotion regulation strategy and intolerance of uncertainty, our ability to anticipate how individuals will perform in changing contexts may be limited. The observed partial or complete transfer effects might mask substantial variability in how different individuals respond to the switch in context. This variability could lead to misleading generalizations, where the model captures average trends but overlooks distinct behavioral patterns in subgroups. Consequently, the predictive model may only partially explain how performance and efficiency are expected to evolve, underscoring the need to integrate individual differences for a more accurate prediction of responses to changing environments.

Accordingly, we administered multiple emotion-cognition traits, including the Emotion Regulation Questionnaire (ERQ[63]), the Intolerance of Uncertainty Scale-12 (IUS-12[61]), and the Behavioral Inhibition and Activation Scales (BIS-BAS[62]) to capture individual differences in emotion regulation strategies, including emotion suppression and cognitive reappraisal, beliefs and emotional reactions related to uncertainty, and sensitivity to punishment and reward, respectively.

We aimed to expand on our transfer models by incorporating information from these questionnaires on individual differences in emotion-cognition traits. To this end, we conducted two separate linear mixed effects models predicting performance and efficiency over the final two rounds of the learning task. Round was included as a within-subjects fixed effect, and stress condition, task version, switching, and all emotion-cognition measures were included as between-subjects fixed effects. All two-, three-, and four-way interactions were included as fixed effects.

In the performance score model, we observed a significant negative interaction between switching and cognitive reappraisal scores on the ERQ, such that participants who tended to use cognitive reappraisal more in their daily life also tended to perform worse in the switching group ($\beta = -0.578$, 95% CI [$-0.984$, $-0.171$], $p = 0.005$; supporting H3). This was particularly true for participants who switched from the consistent to the inconsistent context, as indicated by a Cognitive Reappraisal × Training Version × Switching interaction ($\beta = 0.62$, 95% CI [0.054, 1.186], $p = 0.032$; supporting H3). Visual inspection of the data showed that this effect was driven by a difference in Round 5 performance scores, with low cognitive reappraisers

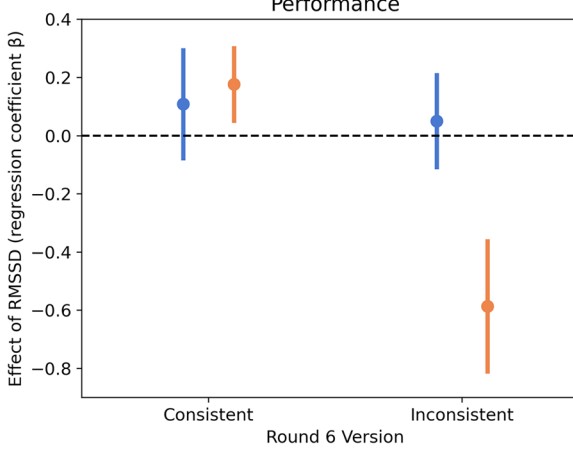
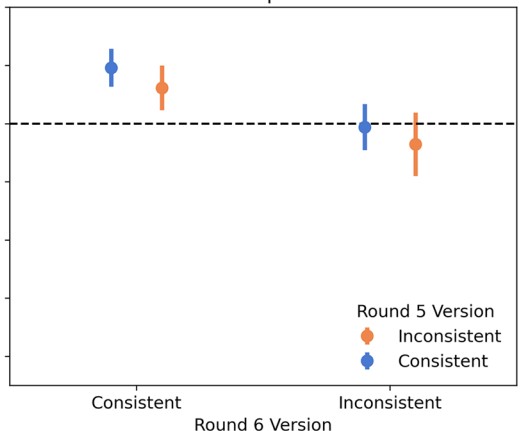
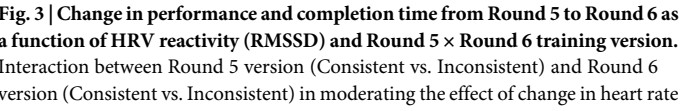

**Fig. 3 | Change in performance and completion time from Round 5 to Round 6 as a function of HRV reactivity (RMSSD) and Round 5 × Round 6 training version.** Interaction between Round 5 version (Consistent vs. Inconsistent) and Round 6 version (Consistent vs. Inconsistent) in moderating the effect of change in heart rate variability (RMSSD) from baseline on change in (left) performance score and (right) time to complete the round from Round 5 to Round 6. Error bars represent ±1 SE. $n = 232$ participants.

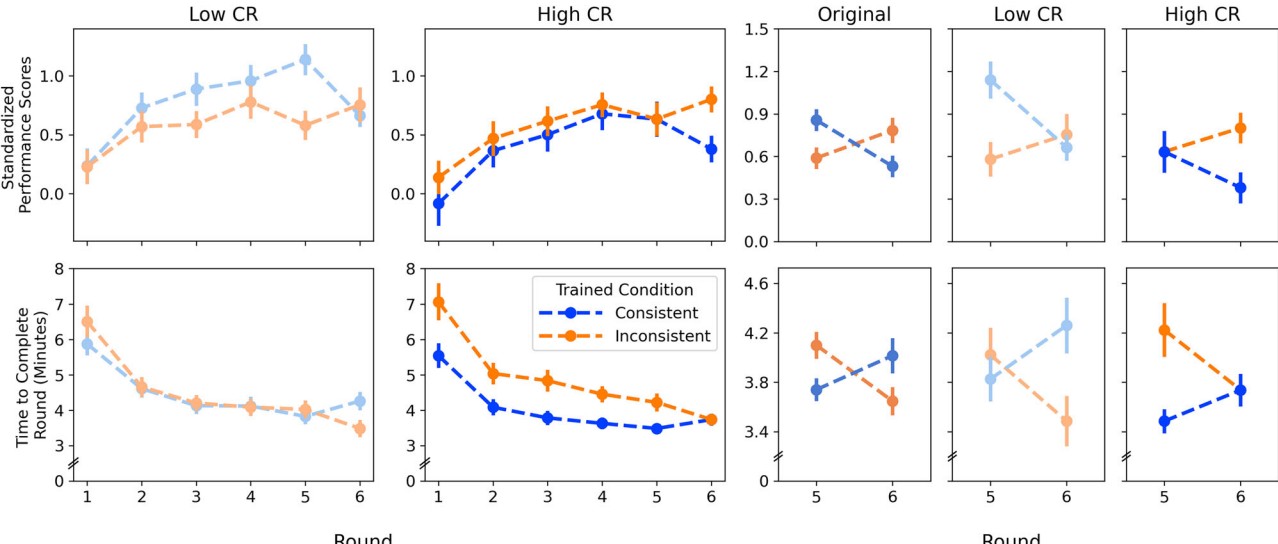

**Fig. 4 | Switchers' performance and completion-time trajectories by cognitive reappraisal (CR) reveal trait-dependent convergence/divergence masked at the group level.** Blue lines represent data from participants who trained in the consistent version, then switched to the inconsistent version. Orange lines represent data from participants who trained in the inconsistent version, then switched to the consistent version. Standardized performance scores are in the top row, completion times are in the bottom row. Left two columns: Separated by trait cognitive reappraisal (CR) levels (low CR in muted hues, high CR in bright hues) across the trained task version (Rounds 1–5) and switch to the other version (Round 6). CR is mean split only for visualization purposes. Right three columns: Rounds 5 (final round of trained version) and 6 (untrained version). The third from the right column (medium hues) represents all switchers regardless of CR level. The rightmost two columns represent those low in CR (muted hues) and high in CR (bright hues). The right three columns demonstrate that group level results misrepresent performance of the individuals within the group: Accounting for CR reveals that what appears as a crossover effect is actually converging or diverging performance depending on individuals' CR. Error bars represent ±1 SE. $N = 123$ participants with Rounds 1–5, $N = 120$ participants with Rounds 5–6.

having substantially better performance at Round 5 than high cognitive reappraisers (Fig. 4). Furthermore, the previously observed transfer effect varies across the spectrum of cognitive reappraisal scores: low cognitive reappraisers tended to perform much better before switching if trained in consistent contexts and equally across both versions after switching, whereas high cognitive reappraisers tended to perform equally across versions before switching and diverged afterwards (Fig. 4).

We also observed interaction effects in our efficiency model. Participants who tended to use more cognitive reappraisal when switching from the inconsistent to consistent version ($\beta = 0.392$, 95% CI [0.021, 0.763], $p = 0.038$; supporting H3) tended to complete their 6th round more slowly. Again, a visual inspection of the data showed that these interactions were driven by a variation in the transfer effect across the cognitive reappraisal spectrum: low cognitive reappraisers tended to complete the task equally as fast across versions before switching and diverged afterwards, whereas high cognitive reappraisers tended to complete the task faster before switching if trained on the consistent version, and converged afterwards (Fig. 4).

Similar interaction effects on performance were observed with intolerance of uncertainty scores on the IUS-12, behavioral inhibition scores on the BIS-BAS, and fun-seeking behavioral activation scores on the BIS-BAS (see Supplementary Tables S9–S13 for full model outputs and Supplementary Fig. 2 for visualizations of transfer effects including individual participant trajectories). Visualizations again showed that transfer effects varied across the spectra of emotion-cognition traits, with post-switch performance divergence occurring for participants with greater intolerance of uncertainty and lower fun-seeking (Fig. 5; supporting H3). Post-switch efficiency diverged for participants with lower intolerance of uncertainty, lower inhibition, and greater fun-seeking.

## Discussion

This study provides a multifaceted exploration of the factors influencing skill transfer, integrating intervention-level uncertainty and stress manipulation, individual differences in emotion-cognition traits and stress responses, and within-person changes in performance with practice and context shifts. The findings integrate within-person practice dynamics with

between-person trait/physiology moderators to explain why group means can mask compensatory adaptations at transfer.

At the intervention level, task stability (consistent vs. inconsistent) during training emerged as a key predictor of performance. Participants trained in consistent environments with stable stimulus-response mappings demonstrated superior efficiency during the acquisition phase compared to those in inconsistent contexts. These results align with prior research emphasizing the role of task structure in promoting skill acquisition and transfer[3,30]. A small, version-specific nonlinearity was also evident: In the inconsistent condition, performance increased from Rounds 1–4, dipped at Round 5, then recovered in Round 6 (Fig. 2C). This Round 5 dip was present only in the inconsistent version but not influenced by switching. We interpret it as a plausible fatigue/effort-allocation effect near the perceived end of training. This may be potentially amplified by the greater demands of the inconsistent mapping, although we lack direct fatigue measures and therefore treat this interpretation cautiously.

However, transfer to untrained versions of tasks revealed asymmetries that provide nuanced insights into the challenges of learning transfer. When participants switched from a consistent to an inconsistent version of the task, both performance and efficiency declined significantly, reflecting the cognitive demands of adapting to novel and less structured environments[18,82]. Conversely, participants switching from an inconsistent to a consistent task version demonstrated more accurate and faster performance.

Individual differences in emotion-cognition traits played a pivotal role in shaping both pre-switch performance and post-switch transfer. Notably, cognitive reappraisal, intolerance of uncertainty, and behavioral inhibition influenced how participants responded to a change in context from their training environments. For example, low cognitive reappraisers excelled in predictable, consistent contexts but struggled to maintain performance scores when transitioning to inconsistent environments, potentially reflecting a reliance on task-specific strategies that did not generalize well. In their case, a reduced ability to engage in cognitive reappraisal may have limited their flexibility, favoring accuracy at the cost of adaptability and speed when faced with novel demands. Behavioral

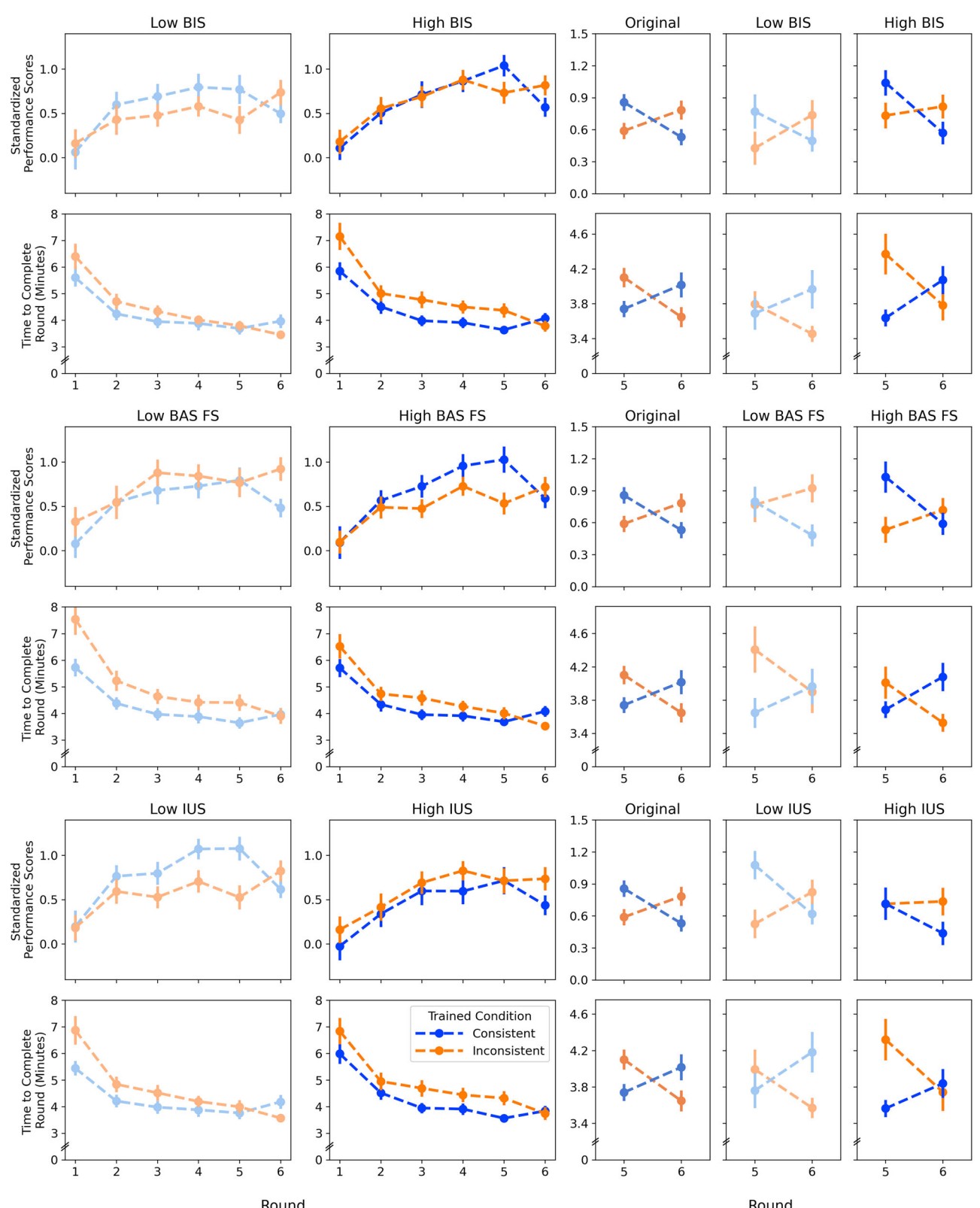

inhibition likewise shaped transfer: participants lower in behavioral inhibition showed greater divergence in efficiency after the switch, whereas those higher in behavioral inhibition exhibited more conservative speed adjustments, consistent with avoidance sensitivity constraining rapid reconfiguration.

Conversely, participants with higher cognitive reappraisal scores performed similarly in both consistent and inconsistent contexts following

training. However, when switching from inconsistent to consistent contexts, their performance dropped, potentially due to over-reliance on specific strategies developed during training. Despite those performance decrements, these individuals demonstrated faster task completion post-switch, suggesting that their ability to flexibly adjust their emotional responses in a changing environment contributed to improved efficiency. This pattern highlights a possible speed-accuracy tradeoff, where reappraisers prioritized

**Fig. 5 | Switchers' performance and completion-time trajectories by IUS, BIS, and BAS Fun-Seeking (BAS FS) show emotion–cognition traits reshape apparent "crossover" effects in Round 6.** Data from participants who switched versions at Round 6 separated by intolerance of uncertainty (IUS; top two rows, behavioral inhibition system sensitivity (BIS: middle two rows), and behavioral activation system sensitivity-fun seeking (BAS FS; bottom two rows). Blue lines represent data from participants who trained in the consistent version, then switched to the inconsistent version. Orange lines represent data from participants who trained in the inconsistent version, then switched to the consistent version. The upper row in each set of results shows standardized performance scores. The lower row in each set of results shows completion times. Left two columns: Separated by low and high levels of each trait (low levels in muted hues; high levels in bright hues) across the trained task version (Rounds 1–5) and switch to the other version (Round 6). Mean split for visualization only. Right three columns: Rounds 5 (final round of trained version) and 6 (untrained version). The third from the right column (medium hues) represents all switchers regardless of IUS/BIS/BAS FS level. The rightmost two columns represent those low in IUS/BIS/BAS FS level (muted hues) and high in those traits (bright hues). The right three columns demonstrate that group level results misrepresent performance of the individuals within the group: Accounting for individual differences in emotion-cognition traits reveals that what appears as a crossover effect is actually converging or diverging performance depending on individuals' IUS, BIS, and BAS FS. Error bars represent ±1 SE. $N$ = 123 participants with Rounds 1–5, $N$ = 120 participants with Rounds 5–6.

faster task completion at the expense of initial accuracy as they adapted to the new context.

Differences in Round 5 performance help explain why some individuals converge and others diverge in performance after switching training contexts. Emotion-cognition traits influenced how much the initial training context affected participants' performance before the switch. When the context had a strong effect pre-switch, participants were more likely to show divergence post-switch. In contrast, when context had little impact pre-switch, participants tended to converge afterward. This pattern suggests that pre-switch performance levels—not just post-switch learning slopes—are critical for interpreting transfer outcomes. Although post-switch learning rates were similar across groups, individual differences in initial performance led to different transfer trajectories. These findings imply that post-switch performance gaps may reflect compensatory effort rather than poor transfer per se. Moreover, because these compensatory responses were shaped by emotion-cognition traits, interventions may need to focus on tailoring pre-switch performance supports to promote equitable transfer outcomes.

Stress, while not directly affecting group-level performance, introduced significant variability in individual responses. The stress manipulation effectively elevated heart rate and blood pressure, confirming physiological arousal. However, HRV was unaffected by the manipulation. Instead, individual differences in HRV, irrespective of stress condition, were associated with performance and efficiency transfer between task versions. Prior work has shown that trait-like or individual-level variability in HRV is often a stronger predictor of cognitive control and emotion regulation than state-level or condition-driven changes[72,79]. Moreover, Blascovich and Mendes's biopsychosocial model of challenge and threat emphasizes that physiological markers like HRV may reflect not just the presence of stress, but the individual's appraisal and regulatory capacity in responding to it[78]. Thus, though we did not observe evidence for group differences in HRV following stress manipulation, we did observe a predictive role of baseline HRV on performance among individuals.

The heterogeneity in our performance trajectories further underscores the value of idiographic and person-centered methods for designing and evaluating training that is responsive to individual profiles rather than group averages[83]. While group-level trends revealed general patterns of learning transfer, individual differences in emotion-cognition traits demonstrated substantial heterogeneity in responses. Recent person-centered work on transfer motivation and transfer factors likewise shows that distinct trainee profiles differ systematically in the extent to which training is implemented on the job, underscoring that averaging across subgroups can obscure meaningful patterns in transfer[84,85]. Our findings parallel this work by showing that subgroups defined by emotion-cognition traits follow different pre-switch and post-switch trajectories. For example, participants with high intolerance of uncertainty struggled more in inconsistent training contexts and demonstrated poorer transfer outcomes, whereas those with low intolerance exhibited stronger pre-switch performance and faster convergence post-switch. These findings support prior calls for methodologies that prioritize within-person variability to capture individual learning and transfer trajectories accurately[45,46] and extend them by linking trajectories to emotion-cognition and physiological profiles.

## Limitations

Although this study provides valuable insights into the mechanisms underlying skill transfer, several limitations must be acknowledged. First, the learning task used in this study, while effective at manipulating context, may lack the complexity of real-world skill acquisition and application. This limits the generalizability of our findings to real-world scenarios, where contextual, social, and motivational factors may significantly influence transfer outcomes. Additionally, while the task incorporated elements of stress and uncertainty through contextual manipulations, it did not target broader or more varied skill domains, such as multitasking or strategic planning in open-ended environments. Moreover, although the task required some degree of dynamic adaptation as it unfolded over time and conditions changed for some participants, it does not capture the full range of adaptive demands present in real-world learning contexts. Finally, our sampling frame was restricted to younger adults (18–30), which limits generalizability to middle-aged and older adults. Age is associated with systematic changes in autonomic functioning (e.g., heart-rate variability, blood pressure), cognitive control, and emotion regulation, all of which may alter how stress and task structure influence learning and transfer. As a result, the patterns observed here may not fully capture how older adults adapt to changing contexts, and future work could explicitly test age as a moderator of stress reactivity and transfer.

Second, our consistent and inconsistent contexts were not equated in baseline difficulty by design. The inconsistent condition was intended to model volatile stimulus-response mappings that tax monitoring and adaptation. This asymmetry helps test directional transfer. We mitigated memory load with an always-visible mapping legend, yet future studies aiming to equalize baseline difficulty could use practice-to-criterion before transfer or pre-test–based titration to match initial accuracy across contexts prior to the switch.

Third, while the stress manipulation was effective in eliciting physiological responses, it may fall short of capturing the complexity of real-word stressors in naturalistic settings, such as workplace demands or academic pressures. The relatively short duration of acute stress exposure limits our ability to assess longer-term adaptation processes that likely shape skill transfer over time. Moreover, while our physiological measures (e.g., HRV, blood pressure) provide meaningful insight into autonomic reactivity, future research incorporating additional biomarkers, such as cortisol, impedance cardiography, or galvanic skin response, could yield a more comprehensive profile of stress-related changes. Although the MAST reliably recruits the HPA axis, we did not assay HPA-axis hormones (e.g., salivary cortisol) and therefore cannot directly characterize endocrine stress responses in this study.

Fourth, while we integrated individual differences in emotion-cognition traits, such as cognitive reappraisal and intolerance of uncertainty, this approach remains incomplete. Other potentially critical factors, such as prior learning experiences, intrinsic motivation, working memory capacity, and domain-specific expertise, were not assessed. These unmeasured variables may interact with both training context and individual traits, potentially explaining additional variance in transfer outcomes. For example, individuals with extensive prior experience in strategy-based tasks may exhibit different learning trajectories and transfer patterns compared to novices.

Finally, although we demonstrated the critical role of Round 5 performance in shaping post-switch convergence or divergence, the

mechanisms underlying this relationship remain unclear. It is uncertain whether pre-switch performance differences stem primarily from differences in training engagement, emotional responses, or cognitive adaptability, and further research is needed to disentangle these factors.

## Future directions

To enhance the ecological validity of skill acquisition and transfer research, future studies can incorporate more complex, real-world tasks that involve collaboration and decision-making under uncertainty. Such tasks can provide deeper insights into how skills transfer across diverse contexts[8]. Additionally, long-term longitudinal studies are essential to examine how transfer mechanisms evolve over time, assessing whether individuals develop greater adaptability with repeated exposure to varied training conditions[1]. Incorporating physiological measures of stress, such as cortisol levels and galvanic skin responses, can offer richer insights into the neural underpinnings of stress and its effects on transfer[86]. Assessing the impact of chronic stress or stress resilience on learning and transfer over time could inform interventions that promote long-term adaptability[87].

Expanding the assessment of individual differences is crucial as they significantly impact learning and transfer outcomes[20]. Exploring interactions between task characteristics and individual differences can provide a more holistic understanding of how personal attributes shape transfer efficiency[88]. Predictive modeling, integrating data on stable traits, situational states, and training context, could identify optimal training strategies for individual learners, enabling personalized training programs tailored to individual needs[89]. Finally, exploring interventions that enhance pre-switch performance, which appeared to determine post-switch convergence or divergence, could maximize transfer efficiency and reduce variability in outcomes[90].

## Implications

Our results suggest a two-armed strategy for designing training that transfers: tune the environment and tailor to the person. On the environment side, introducing structured variability (e.g., controlled re-mapping of stimulus–response rules, deliberate mid-course switches) may slow early learning but prepare learners for changes at deployment. On the person side, brief assessments of emotion-cognition traits (e.g., intolerance of uncertainty, cognitive reappraisal) and feasible physiological indices of adaptive regulation (e.g., HRV change where instrumentation is available) may inform adaptive scaffolding, such as providing advance cues and rule legends for high-uncertainty learners, reappraisal coaching for learners low in cognitive reappraisal, and extra practice in volatile contexts for those who show limited adaptive regulation. For example, placing learners whose emotion-cognition profiles favor routine stability into roles or modules emphasizing consistent mappings, or using variability-rich practice to cultivate flexibility in those slated for volatile environments.

The asymmetries we observe, including performance disruption when moving from consistent to inconsistent contexts, alongside rapid convergence in efficiency, are consistent with a dual-systems account in which model-free routines optimize stable contexts, while model-based control supports reconfiguration under volatility at a cost. Our trait and physiology results suggest that emotion regulation, uncertainty tolerance, and HRV change moderate flexibility between these two strategies. Integrating intervention-level determinants with these person-level moderators helps explain why group means can mask compensatory adaptations at transfer and yields concrete predictions. That is, transfer success depends jointly on whether the change requires remapping stimulus–response rules or redefining the task objective, on practice history, and on trait profiles that govern willingness and capacity to re-plan. By formalizing these interactions, the present work extends structural accounts of transfer to a multi-level framework in which environment design, internal control architecture, and emotion-cognition dynamics determine when and how skills generalize.

By measuring objective performance, efficiency, and autonomic responses, we demonstrate how group-level transfer effects can mask compensatory adaptations between individuals. This multi-level, person-centered approach extends existing transfer theories and offers a framework for designing training environments that are both structurally robust and responsive to individual profiles. Together, our findings suggest that transfer of learning may be jointly shaped by task structure, stress context, and individual differences in emotion-cognition and physiological regulation.

## Data availability

Deidentified data are posted on Open Science Framework (OSF) at https://osf.io/x72z3. All data needed to evaluate the conclusions in the paper are present on OSF.

## Code availability

A code notebook and data analysis scripts are posted on Open Science Framework (OSF) at https://osf.io/x72z3.

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

## Acknowledgements

This work was sponsored by the U.S. Army Research Institute for the Behavioral and Social Sciences (ARI) and was accomplished under Grant # W911NF-22-1-0238. The views, opinions, and/or findings contained in this report (paper) are those of the authors and shall not be construed as an official Department of the Army position, policy, or decision, unless so designated by other documents. The funders had no role in study design, data collection and analysis, decision to publish or preparation of the manuscript. The authors would like to thank the following research assistants for their invaluable contributions to data collection and participant coordination: Mariel Barnett, Elyssa Barrick, Mallory Bouque, Kat Hradek, Lauren Himmel, Emilio Izquierdo, VeAnn Lee, Essence Leslie, Carly Lubowe, Ethan Lopykinski, Michelle Orioha, Radha Patel, Madison Peebles, Elise Rolston, Geetha Thomas, Allie Valocchi, Jaylene Vázquez, and Isabella Zolikoff.

## Author contributions

Design and Conceptualization of the Broader Funded Project: K.J.L., D.J.F., A.P.B., & B.N.M.; Study conception and design: K.J.L. & B.N.M.; Analysis and Interpretation of Results: K.J.L. & D.J.F.; Draft Manuscript Preparation: K.J.L. All authors reviewed the results and approve of the final version of the manuscript.

## Competing interests

The authors declare no competing interests.
