## [Transparent Peer Review file · Communications Psychology]

Task, person, and experiential characteristics drive the transfer of learning

Corresponding Author: Dr Kyle LaFollette

Version 0:

Decision Letter:

Dear Dr LaFollette,

Thank you for your patience during the peer-review process. Your manuscript titled "Task, person, and experiential characteristics drive the transfer of learning" has now been seen by 3 reviewers, and I include their comments at the end of this message. They find your work of interest but raised some important points. We are interested in the possibility of publishing your study in Communications Psychology, but would like to consider your responses to these concerns and assess a revised manuscript before we make a final decision on publication.

We therefore invite you to revise and resubmit your manuscript, along with a point-by-point response to the reviewers. Please highlight all changes in the manuscript text file.

Editorially, we consider it important that the revised manuscript includes an improved literature review that supports the derivation of hypotheses. The Methods require greater clarification and rationale for choices, especially regarding the modeling choices and power. Finally, the Discussion needs to better integrate the findings into the existing literature.

I am attaching an Editorial Requests Table that details critical reporting requirements for the revised manuscript. Please attend to each item and ensure your manuscript is fully compliant. If your revised manuscript is not aligned with these requests on major issues, such as those concerning statistics, it may be returned to you for further revisions without re-review.

Please submit the following items:

- Revised manuscript
- Point-by-point response to the referees' comments
- Cover letter (as a separate document)
- <https://www.nature.com/documents/nr-reporting-summary.pdf>>Nature Research Reporting Summary
- Completed Editorial Request Table (attached).

via this link: Link Redacted .

Additional guidance is available in our style and formatting guide Communications Psychology formatting guide.

Best regards,

Jennifer Bellingtier

Jennifer Bellingtier, PhD
Senior Editor
Communications Psychology

Dr Erdem Pulcu
Editorial board member
Communications Psychology
0000-0002-2170-0677

REVIEWER EXPERTISE:

Reviewer #1 learning, games
Reviewer #2 learning, transfer
Reviewer #3 learning, physiological reactivity

REVIEWER REPORTS:

Reviewer #1 (Remarks to the Author):

I have several comments/questions below, but I'd like to preface by saying that I thought the paper was overall well done. I think it's an interesting question – one that isn't considered enough in many domains and thus is very much worth considering and having more work on – and I think that the data and discussion will be valuable to the literature. Given that in academia we tend to focus on our quibbles and not our points of agreement (e.g., as shown by the fact that my questions and comments are three pages long and this introductory paragraph is three sentences long), I at least wanted that up front.

As for questions/comments:

Introduction:

1) I had some difficulty figuring out the patterns of citations as they seemed a bit idiosyncratic to me in terms of which domains were/were not considered or used as foundational to the main questions and hypotheses.

For instance, the authors write, "researchers may operationalize context as the physical environment when examining learning transfer" and for this statement they exclusively cite context-dependent memory papers. But in other cases, the authors are discussing motor or cognitive learning using nearly identical phrasing. Personally I'm not sure I would necessarily put those all in the exact same bin with regard to what it means to transfer learning, but perhaps the authors would. However, I think it does readers who might not know the field very well a disservice to not have this called out in any fashion (e.g., being more explicit about what domain the work that is being referenced is situated within would allow readers to know how far of an extrapolation from existing work the current work is – e.g., how strong a case can be made from underwater list learning to gameplay under uncertainty).

Furthermore, given the task that is used, it's unusual how little (if at all) the pretty broad literature on decision-making under uncertainty is referenced. There is obviously quite a lot of work on how individuals change their decision-making behaviors as they switch between decision-making environments and what types of environments/learning strategies promote the most flexible behaviors (i.e., most transferable learning). The big literature on what types of tasks allow for effective model-based versus model-free behavior, for instance, speaks directly to some of these points.

As a much more minor point, I thought it was unusual for von Bastian and colleagues to be called out by name. Perhaps it's

the use of the word “acknowledge” that seemed peculiar. But I’m not sure that anyone in the entire field believes that learning transfer can be “fully explained by cognitive traits or the quality of training alone.”

2) It might be worth discussing the fact that differences in the intervention level is literally the basis for full fields and is arguably the primary focus of study in others. For instance, within the field of perceptual learning there are myriad papers focused entirely on the question of what manipulations of task produce changes in the extent to which learning generalizes. That’s arguably the main thrust in much of the cognitive training literature as well. And the reason I think that’s important is that it’s in contrast to the much lower focus on between-person differences. The narrative right now puts those at a similar level in terms of what the field has considered and knows about and it’s not clear to me that’s the case. I’d say we know *much* more about how task characteristics influence generalization as compared to what we know about individual difference factors (e.g., there is work in basically every learning domain regarding how variability during training impacts initial learning and then subsequent generalization: DOI: 10.1016/j.tics.2022.03.007 and that’s true for a variety of similar broad principles; but there’s far less on trait-level characteristics).

Methods:

I’m not sure if this fits best within methods or discussion, but I found it surprising that the authors chose to utilize a training and transfer task with such drastically different base difficulties. Obviously it’s possible to deal with that (to some degree) analytically, but it runs counter to the approach in many domains where researchers will use training and transfer tasks that have reasonably equivalent difficulty (e.g., in perceptual research using discriminations around 45 degrees for training and 135 degrees for generalization; or in cognitive – block span for training and digit span for transfer; or in motor research one type of perturbation for training and another equivalently difficult perturbation for generalization). So it might be useful for the authors to discuss this choice either in setting up the methods or the results and how future researchers should think about that (sometimes it can’t be avoided).

Results:

1) Obviously the performance data through time was non-linear and so it would be necessary to include something to deal with that fact in the analysis if the goal was to do GLM stats. However, why was the particular form chosen (round squared). That form seems to violate what we know about human learning and I’m not sure it’s actually a good fit to the data (most human learning data of this sort at the individual level is best fit by something from the exponential family/exponential to asymptote). Given that so many of the inferences are drawn from the results using that particular parameterization, it would be useful to know why it was chosen (i.e., if there’s theory or empirical reasons to pick it) and how the results might shift if a (potentially) better fitting form was chosen instead.

<https://pubmed.ncbi.nlm.nih.gov/10909131/>

<https://pubmed.ncbi.nlm.nih.gov/17576267/>

<https://doi.org/10.1167/jov.21.13.5>

2) There’s a peculiar drop in performance for the inconsistent group during round 5 (one that doesn’t seem like it could be due to random chance – it’s a huge deviation from what would be expected given the trend from rounds 1-4 and the sample size). Do the authors have a reason for that? There doesn’t seem to be any change in the size of the error bars for that point, which I find surprising (I would have guessed that it was due to some participants fatiguing or something like that, but that would tend to spread the distribution out and thus cause broader error bars).

3) Given that the focus was on individual differences, it’s a bit disappointing to not have any individual difference plots. It’s honestly a little dissatisfying to only have aggregated plots when the focus is on how people differ (it would also be useful for readers to be able to see what some of the individual difference level distributions look like rather than just model outputs).

Reviewer #2 (Remarks to the Author):

Task, person, and experiential characteristics drive the transfer of learning

General

1. Thank you for granting me the opportunity to review this manuscript. I think the manuscript has adopted an interesting experimental design and some valuable measures. Also the data-analysis seems to have been done in a solid manner. This makes this paper potentially valuable for both research and practice.

However, I have serious concerns with regard to the theoretical foundation of the paper. The current version lacks a clear line of argumentation for the relevance of the manuscript and a theoretical lens used to conceptualize the research questions. Also the substantiation for why specific choices were made is described only limitedly. For example, why were specific predictors chosen? Which theory or empirical findings back up these choices? And why are the employed measures a good alternative for self-reported measures? Finally, there is a lack of structure in the manuscript as specific information described in for example the results section needs to be described in the method section, which also applies for information presented in the present study section.

In general, I think a considerable improvement in structure, argumentation and theory is required to make this manuscript publishable. Below, more detailed comments per section are provided.

Theoretical framework

2. The theoretical framework is brief at its current form. It lacks a clear theoretical lens based on which a research rationale has been worked out. Also different concepts could be explained more thoroughly (i.e., definitions of concepts) and how they relate to one another. I also miss a clear definition of transfer. What is your definition of transfer? And how is this operationalized?
3. At the beginning of the introduction you state that there is disagreement among researchers in what constitutes transfer and which predictors should be included. Based on the current version, it is unclear how the approach suggested by you will be a step forward in this discussion, nor how your study would be a contribution for practitioners and/or trainers. This makes it difficult for me as a reader to understand what the utility is of your study.
4. Even though I acknowledge your approach towards pre-registering your hypotheses, it is still very important for the reader to understand why you expect specific findings. Therefore, I would recommend working out a rationale for why your formulated specific hypotheses and to back up those claims by means of different empirical findings and/or theory.
5. Specific elements described in the present study would fit better in a method section. Especially regarding the sample and measurement instruments.
6. It is potentially interesting that you use more objective measures such as blood pressure and electrocardiograms. This is a step forward in comparison to self-reported measures. Nevertheless, choices for specific types of measures should be justified more clearly with theory. Information about its validity and previous studies using these methods is important as this can help the reader understand why your approach is valid and reliable. This also applies for the intervention manipulations you mention in the present study section. It is unclear why these manipulations were made and whether they were used in previous studies.

Method

7. I would recommend to first describe your methodology and then to move on towards the results and discussion section. This would also increase the likelihood that the right information is presented in the right section. I now read information related to the methods in the results section (see first alinea until at least manipulation checks) and in the present study section. In general it would make it easier for the reader to understand why you made specific choices.
8. A context description of your study would be of help to understand why a game setting has been used, and how this relates to real life tasks of your sample (or not). This is especially important as you also mention this as a limitation in your discussion section.
9. Information about the reliability and validity of the used questionnaires should be presented in the method section and not in the result section.
10. The procedure section seems to be worked out comprehensively. Although I must also admit that I am not an expert on the used procedure. Perhaps other reviewers provide valuable feedback on this.
11. I would recommend describing a data-analysis section in which you explain your statistical approach towards analyzing the data.

Results

12. The result section provides a lot of information and details about the analyses done to investigate your research aims. For the reader, it would help if you could describe how different statistical analyses serve the research purposes. Why do you perform specific analyses? And which research questions are answered by doing this? In the current version there are no clear research questions, whereas this would support you in working out a comprehensive structure in your results section. Another option is to do so based on your hypotheses, to follow that structure in your result section and to briefly mention each hypothesis (or several) per analysis.

Discussion

13. In the theoretical framework you mention that you look at both within- and between-level effects. However, this is not mentioned again at the beginning of the discussion section as you only mention the within-level there. Does this imply that you have not looked at the between-person level? In any case, consistency is warranted here.
14. I also miss linkages with the formulated hypotheses. To what extent were your findings in line with your expectations and for which hypotheses did you not find support? How does it relate to previous studies? And how could unexpected findings be explained?
15. I would recommend working out a separate paragraph in which you describe your the theoretical implications of your study. You now do describe some contributions of your study in the last alinea of the discussion, but this is still relatively limited. This would support the rationale for why your study has value.
16. In addition to that, an additional paragraph in which you describe the practical implications of your study in a more concrete manner would be of value as well. In the current version, the practical implications are only described superficially which hampers the value of your study for practice.

Reviewer #3 (Remarks to the Author):

I want to sincerely thank the authors for their valuable work. The authors present a meaningful and timely investigation into the role of individual emotion-cognition traits in learning transfer under stress, which a topic with clear relevance for both psychological theory and practical training applications. The use of context switching and stress manipulation in a learning task, combined with physiological and self-report measures, offers a valuable contribution to the growing body of work on how affective and cognitive systems interact during complex learning processes. I personally think the manuscript would benefit from several important revisions to improve clarity, structure, and theoretical framing. The literature review could more

clearly articulate the empirical and theoretical gaps the study addresses. Additionally, key design elements—such as task environments, conditions, and individual difference measures—need more background and justification. And the Discussion would be strengthened by more explicitly connecting the findings to theoretical frameworks and clarifying how this work advances the field.

The majors and minor comments are provided below:

Major comments:

1. The structure and organization of the manuscript, particularly the Methods and Results sections, require revision for clarity and coherence. First, I am not sure why the Method section is the last main section of the manuscript (after the Discussion section). Is it an accident? It is quite confusing to read without knowing the detail of the measures and tasks. Second, a substantial portion of content currently placed at the beginning of the Results section (up to the “Manipulation Check” heading) describes the task and experimental procedures and would be more appropriately located in the Methods section. To improve readability and align with standard reporting practices, I recommend reorganizing the manuscript so that all task descriptions, measures, and procedures are clearly presented within the Methods section, followed by the Results.

2. You mentioned that “physiological stress responses were measured using electrocardiogram, heart rate, and blood pressure”. While these are valuable indicators of autonomic activity, particularly sympathetic nervous system (SNS) activation, it is important to note that cold pressor tasks also robustly engage the hypothalamic-pituitary-adrenal (HPA) axis. Prior research has frequently included salivary cortisol (as you noted in the limitation section) as a key biomarker of HPA axis activation and a reliable measure of stress response in addition to blood pressure. (For example: Liu et al., 2024; Minkley et al., 2014; Schwabe et al., 2008, Schwabe & Schächinger, 2018, etc.) Including relevant citations would help justify the use of these measures (i.e., blood pressure, HRV, heart rate) and clarify how they reflect stress reactivity.

Liu, Y., Byrne, K. A., Aly, H., Ghaiomy Anaraky, R., & Knijnenburg, B. (2024). Can stress put digital privacy at risk? Evidence from a controlled experiment examining the impact of acute stress on privacy decisions on a simulated social network site. *Cyberpsychology, Behavior, and Social Networking*, 27(9), 664-672.

Minkley, N., Schröder, T. P., Wolf, O. T., & Kirchner, W. H. (2014). The socially evaluated cold-pressor test (SECPT) for groups: Effects of repeated administration of a combined physiological and psychological stressor. *Psychoneuroendocrinology*, 45, 119-127.

Schwabe, L., Haddad, L., & Schächinger, H. (2008). HPA axis activation by a socially evaluated cold-pressor test. *Psychoneuroendocrinology*, 33(6), 890-895.

Schwabe, L., & Schächinger, H. (2018). Ten years of research with the Socially Evaluated Cold Pressor Test: Data from the past and guidelines for the future. *Psychoneuroendocrinology*, 92, 155-161.

3. The introduction provides a broad overview of learning transfer and individual differences, but it would benefit from a more clearly articulated theoretical or empirical gap. While the literature review outlines relevant prior work, it does not sufficiently specify what existing studies have found, where they fall short, or how this manuscript addresses those limitations. The authors could explicitly highlight more on what is currently missing in the literature and clearly state how their study contributes novel insights or methodological advancements beyond prior research.

4. Is there a specific reason that you only recruited participants from 18 - 30 years old? Age is a well-established factor influencing cognitive abilities, emotional regulation, and learning strategies—all of which are central to the current study. Limiting the sample to younger adults may restrict the generalizability of the findings, particularly given that older and middle-aged adults often differ meaningfully in the psychological constructs under investigation.

5. A more detailed explanation of key experimental variables including “Consistent vs. Inconsistent” task environments and the “Stress vs. Non-stress” conditions in the introduction section. These factors appear to play a central role in the study design and theoretical framework, yet their relevance is not fully explained. The authors should elaborate on why these variables were selected, how they relate to learning transfer, and what prior research suggests about their potential effects.

6. Did you conduct a power analysis to determine the sample size for each condition?

7. I might have missed something here “Round and round squared were included as within-subjects fixed effects...” Why round 2 was included as predictor in the model?

8. In Figure 2C, I noticed an interesting performance trajectory in the inconsistent condition: performance increases steadily from Round 1 to Round 4, then drops noticeably in Round 5, before recovering again in Round 6. This non-linear pattern, especially the decline in Round 5, stands out and may warrant further discussion. Could you further clarify or provide explanation to this? Does this dip corresponds with a specific task manipulation (e.g., switching context) or participant-level factors (e.g., fatigue, adjustment period)?

9. You mentioned that in the discussion “cognitive reappraisal, intolerance of uncertainty, and behavioral inhibition

influenced how participants responded to a change in context from their training environments.” However, the effect of behavioral inhibition was not discussed.

10. The manuscript would benefit from a more structured and explicit discussion of both the practical and theoretical implications of the findings. For example, the last paragraph under the “Future Direction” section (“Our findings suggest the importance of designing training interventions...”) talks about the implication of this work, including adaptive training, emotion regulation coaching, and personalized scaffolding. I recommend adding a dedicated “Implications” subsection within the Discussion that distinguishes: (1) Practical implications, especially how the findings may inform education, workforce training, or personalized learning interventions based on emotion-cognition traits; and (2) Theoretical implications, specifically how this study advances existing models of learning transfer by incorporating emotion-cognition traits (e.g., cognitive reappraisal, intolerance of uncertainty), and how it addresses empirical or conceptual gaps in the literature. A clearer articulation of the contribution to the literature would enhance the theoretical significance.

Minor Comments:

1. The introduction section did not have heading “Introduction”.
2. In page 4, authors stated that “For example, in a probabilistic learning task, highly anxious individuals may exhibit heightened threat anticipation and overestimate the likelihood of negative outcomes, leading them to rely on rigid, well-learned strategies rather than adapting flexibly to new contingencies.” Is there any scientific evidences supporting this? (Need references)
3. In page 6, the sentence “At the intervention-level we manipulated task predictability (consistent vs. inconsistent) during training...” A comma is missing after “At the intervention-level”.
4. It would be easier to follow if you list the hypothesis as H1, H2, H3, etc... and in the result section, indicating whether each hypothesis was supported.
5. The first sentence in the method section “Half the participants were either recruited from the university’s SONA subject pool, the other half from the local Cleveland community, in exchange for course credit or \$80 USD...” The word “either” should be removed.
6. All “ β ” sign should be italicized.

I sincerely appreciate the authors’ efforts and contributions to this important area of research and wish them the very best in their future work.

Communications Psychology is committed to improving transparency in authorship. As part of our efforts in this direction, we are now requesting that all authors identified as ‘corresponding author’ create and link their Open Researcher and Contributor Identifier (ORCID) with their account on the Manuscript Tracking System prior to acceptance. ORCID helps the scientific community achieve unambiguous attribution of all scholarly contributions. You can create and link your ORCID from the home page of the Manuscript Tracking System by clicking on ‘Modify my Springer Nature account’ and following the instructions in the link below. Please also inform all co-authors that they can add their ORCIDs to their accounts and that they must do so prior to acceptance.

Version 1:

Decision Letter:

Dear Dr LaFollette,

Your manuscript titled "Task, person, and experiential characteristics drive the transfer of learning" has now been seen by our reviewers, whose comments appear below. In light of their advice I am delighted to say that we are happy, in principle, to publish a suitably revised version in Communications Psychology.

We therefore invite you to revise your paper one last time to address the remaining concerns of our reviewers and a list of editorial requests. At the same time we ask that you edit your manuscript to comply with our format requirements and to maximise the accessibility and therefore the impact of your work.

EDITORIAL REQUESTS:

SUBMISSION INFORMATION:

OPEN ACCESS:

* DATA AVAILABILITY:

Link Redacted

Best regards,

Jennifer Bellingtier

Jennifer Bellingtier, PhD
Senior Editor
Communications Psychology

Dr Erdem Pulcu
Editorial board member
Communications Psychology
0000-0002-2170-0677

REVIEWER EXPERTISE:

Reviewer #1 learning, games
Reviewer #2 learning, transfer
Reviewer #3 learning, physiological reactivity

REVIEWERS' COMMENTS:

Reviewer #1 (Remarks to the Author):

The authors were incredibly responsive to my previous comments. All of the changes that were made in text fully addressed the given points and the responses to analyses (e.g., functional form) or presentation (e.g., individual difference plots) were reasonable and compelling. I have no further questions or comments. I think the work is really interesting and should make an excellent contribution to the literature.

Reviewer #2 (Remarks to the Author):

General

1. Thank you for the alterations made in the manuscript. I think it greatly improved the flow the study and the argumentation. Below I provide a number of additional comments.

Theoretical framework

2. In the introduction, you mention that there is a lack of focus on interactions between variables in predicting transfer. I think it would be good to explain a bit more why this is problematic as this argumentation is currently lacking. To be able to substantiate this argumentation, I would recommend the manuscript of Blume et al. (2019) in which an interactionist perspective is also explained including potential contributions of adopting such an approach.

3. In the present study section, you do mention some contributions for practice based on your study, but this could be made more concrete. Especially as to how practitioners can use insights of your study for training or work practice.

Method

4. It is good that you provide more explanation for using more objective measures in the method section. But I think it is also in itself a valuable contribution to the transfer literature. Perhaps it would be good to add it as a contribution of your study in the theoretical framework.

Results

5. No further comments regarding the results section.

Discussion

6. In the discussion section, you discuss the need for person-centered analyses in understanding transfer. Perhaps a study done by De Jong et al. (2023) and/or Quesada-Pallares et al. (2022) could be interesting as these study also employed a person-centered approach towards transfer. Especially with regard to how findings in your study relate to their findings.

7. In the limitation section, you mention that the age range might limit the generalizability of your findings. Here you could provide some more explanation as to why this limits your findings towards a broader population.

8. You have now more clearly worked out implications for both theory and practice. However, I think that these implications could be described in a more structured manner. In the future direction paragraph, the last alinea is more a practical implication, whereas in the implication paragraph you also mention direction for future research (first alinea). Making a clearer distinction is recommendable.

9. I would recommend to end your manuscript with a conclusion paragraph in which you sum up the main contributions of your study.

References

Blume, B. D., Ford, J. K., Surface, E. A., & Olenick, J. (2019). A dynamic model of training transfer. *Human Resource Management Review*, 29(2), 270-283.

De Jong, B., Jansen in de Wal, J., Cornelissen, F., & Peetsma, T. (2023). Investigating transfer motivation profiles, Their antecedents and transfer of training. *Education Sciences*, 13(12), 1232.

Quesada-Pallarès, C., González-Ortiz-de-Zárate, A., Pineda-Herrero, P., & Cascallar, E. (2022). Intention to transfer and transfer following eLearning in Spain. *Vocations and Learning*, 15(2), 359-385.

Reviewer #3 (Remarks to the Author):

Dear Authors,

Thank you for your careful and comprehensive revision. I have no further concerns or suggestions. Congratulations on this valuable contribution, and best wishes for your continued research.

Editor

E1. Editorially, we consider it important that the revised manuscript includes an improved literature review that supports the derivation of hypotheses. The Methods require greater clarification and rationale for choices, especially regarding the modeling choices and power. Finally, the Discussion needs to better integrate the findings into the existing literature.

Thank you for your feedback on this manuscript. In this revision we (1) expanded and reorganized the literature review to more clearly explain the motivation for our hypotheses, (2) clarified Methods with rationale for modeling choices (polynomial vs. exponential/power checks), preprocessing, etc., and (3) rewrote the Discussion to integrate findings with the decision-making-under-uncertainty (model-based/model-free) literature, adding explicit theoretical and practical implications. Related robustness analyses and individual-trajectory visualizations are now provided in the SI.

Reviewer 1

R1.1. I have several comments/questions below, but I'd like to preface by saying that I thought the paper was overall well done. I think it's an interesting question – one that isn't considered enough in many domains and thus is very much worth considering and having more work on – and I think that the data and discussion will be valuable to the literature. Given that in academia we tend to focus on our quibbles and not our points of agreement (e.g., as shown by the fact that my questions and comments are three pages long and this introductory paragraph is three sentences long), I at least wanted that up front.

Thank you for your kind assessment and constructive feedback. Below, we respond point-by-point and have revised the manuscript accordingly to address your concerns.

R1.2. I had some difficulty figuring out the patterns of citations as they seemed a bit idiosyncratic to me in terms of which domains were/ were not considered or used as foundational to the main questions and hypotheses. For instance, the authors write, “researchers may operationalize context as the physical environment when examining learning transfer” and for this statement they exclusively cite context-dependent memory papers. But in other cases, the authors are discussing motor or cognitive learning using nearly identical phrasing. Personally I'm not sure I would necessarily put those all in the exact same bin with regard to what it means to transfer learning, but perhaps the authors would. However, I think it does readers who might not know the field very well a disservice to not have this called out in any fashion (e.g., being more explicit about what domain the work that is being referenced is situated within would allow readers to know how far of an extrapolation from existing work the current work is – e.g., how strong a case can be made from underwater list learning to gameplay under uncertainty).

We agree that transfer effects are studied across distinct literatures (episodic/context-dependent memory; motor/perceptual skill; educational/cognitive training; cognitive control/rule learning) and that we should not imply those effects are interchangeable. We revised the Introduction to (1) name the domain when citing prior work, and (2) broaden the citations in the relevant sentence so readers see which field each example comes from. The passage now reads as follows:

One reason for the lack of agreement is that researchers are inconsistent in how context is operationalized when examining near and far transfer (Barnett & Ceci, 2002). Across domains, context denotes different things. In episodic memory, context often means the physical environment (e.g., underwater vs. on land; context-dependent memory) (Godden & Baddeley, 1975; Murre, 2021; Smith & Vela, 2001). In motor and perceptual skill learning, context typically refers to sensorimotor/task constraints and training settings (e.g., simulation-based surgical skills, sport perceptual training) (Dawe et al., 2014; Hadlow et al., 2018). In educational/cognitive training, context frequently refers to the similarity of stimuli or underlying cognitive operations (near vs. far transfer) (Kassai et al., 2019; Melby-Lervag & Hulme, 2013; Sala & Gobet, 2019; Shipstead et al., 2012; Simons et al., 2016; Spearman, 1904; Thurstone, 1938; Pan & Rickard, 2018). In cognitive control/rule learning, context can denote stimulus–response or rule mappings that vary over time (variably mapped/dynamic tasks) (Macnamara & Frank, 2018; Wen et al., 2023). Across these literatures, ‘transfer’ refers to the extent to which performance and efficiency acquired in a trained context are maintained or appropriately adapted when contextual features change. More recently, a computational and neural framework has been proposed to unify generalization across such contexts (Heald et al., 2023), alongside classic individual-difference perspectives on skill learning (Ackerman, 1987).

R1.3. Furthermore, given the task that is used, it’s unusual how little (if at all) the pretty broad literature on decision-making under uncertainty is referenced. There is obviously quite a lot of work on how individuals change their decision-making behaviors as they switch between decision-making environments and what types of environments/learning strategies promote the most flexible behaviors (i.e., most transferable learning). The big literature on what types of tasks allow for effective model-based versus model-free behavior, for instance, speaks directly to some of these points.

We agree the introduction underplayed the decision-making under uncertainty literature and the model-based/model-free framework that speaks to flexible transfer. We’ve added the paragraph below to the Introduction. This addition clarifies why environments with shifting contingencies (our inconsistent stimulus—response mappings) are expected to recruit model-based control and why stable contexts foster efficient model free routines; it also links stress and individual differences to arbitration between these systems. We have added the following to the Introduction:

A complementary tradition in decision-making under uncertainty distinguishes between model-based (i.e., planning using an internal model or ‘cognitive map’) and model-free (i.e., habitual,

cached-value) control, with critical implications for transfer. Model-based control supports flexible generalization when contingencies change, whereas model-free control yields efficient performance in stable contexts but poorer transfer under revaluation or structural shifts (e.g., goal or transition changes; Daw et al., 2011; Doll, Simon, & Daw, 2012; Behrens et al., 2018). A successor representation stores a predictive map of which situations are likely to come next; it enables partial transfer when goals (rewards) change but the underlying transition structure is preserved (Gershman et al., 2023; Momennejad et al., 2017). People shift between planning and habit when they judge that the benefits of being flexible outweigh the mental effort, and when the task makes planning feasible (Kool, Cushman, & Gershman, 2017; Collins & Frank, 2013).

This literature is far richer on intervention-level manipulations than on trait moderators: across perceptual, motor, and cognitive training, training variability robustly slows initial learning yet improves generalization, establishing task design as a primary driver of transferable learning (Raviv, Lupyan, & Green, 2022). In contrast, we know less about how stable traits and physiological regulation influence this flexibility. Stress, for example, can down-weight model-based control, with working memory buffering the effect (Otto et al., 2013). Taken together, these perspectives predict a pattern of results: training in consistent contexts promotes efficient routines, whereas switching into inconsistent contexts taxes planning/structure learning and reveals individual differences in flexibility.

R1.4. As a much more minor point, I thought it was unusual for von Bastian and colleagues to be called out by name. Perhaps it’s the use of the word “acknowledge” that seemed peculiar. But I’m not sure that anyone in the entire field believes that learning transfer can be “fully explained by cognitive traits or the quality of training alone.”

We agree the phrasing was overly pointed. We’ve removed the named call-out and tempered the claim to reflect the field’s broader view: the magnitude and generality of individual-difference effects and their interactions with training and environment are mixed across studies, motivating a multi-level framing rather than implying consensus against any single-factor account. The passage now reads as follows:

Differing from the longstanding focus on training and cognitive abilities in the learning transfer literature, a recent review (3) suggests that accounts focusing solely on training quality or on learner traits are incomplete. More broadly, the size and generality of individual-difference effects and their interactions with training and environment remain uncertain across studies. Consistent with this view, we frame transfer as the outcome of a dynamic interplay among intervention-level factors, between-person differences, and within-person fluctuations.

R1.5. It might be worth discussing the fact that differences in the intervention level is literally the basis for full fields and is arguably the primary focus of study in others. For instance, within the field of perceptual learning there are myriad papers focused entirely on the question of what manipulations of task produce changes in the extent to which

learning generalizes. That's arguably the main thrust in much of the cognitive training literature as well. And the reason I think that's important is that it's in contrast to the much lower focus on between-person differences. The narrative right now puts those at a similar level in terms of what the field has considered and knows about and it's not clear to me that's the case. I'd say we know *much* more about how task characteristics influence generalization as compared to what we know about individual difference factors (e.g., there is work in basically every learning domain regarding how variability during training impacts initial learning and then subsequent generalization: DOI: 10.1016/j.tics.2022.03.007 and that's true for a variety of similar broad principles; but there's far less on trait-level characteristics).

We agree the literature on intervention-level manipulations, especially variability, schedule, and task structure in perceptual, motor, and cognitive training, has far outpaced work on trait-level predictors. We revised the Introduction to (1) foreground this asymmetry, (2) situate our "uncertainty/variably mapped" framing within that tradition, and (3) clarify that our contribution is testing how traits and physiology interact with well-established task determinants of generalization. We also now cite the TiCS review that you suggested:

A complementary tradition in decision-making under uncertainty distinguishes between model-based (i.e., planning using an internal model or 'cognitive map') and model-free (i.e., habitual, cached-value) control, with critical implications for transfer. Model-based control supports flexible generalization when contingencies change, whereas model-free control yields efficient performance in stable contexts but poorer transfer under revaluation or structural shifts (e.g., goal or transition changes; Daw et al., 2011; Doll, Simon, & Daw, 2012; Behrens et al., 2018). A successor representation stores a predictive map of which situations are likely to come next; it enables partial transfer when goals (rewards) change but the underlying transition structure is preserved (Gershman et al., 2023; Momennejad et al., 2017). People shift between planning and habit when they judge that the benefits of being flexible outweigh the mental effort, and when the task makes planning feasible (Kool, Cushman, & Gershman, 2017; Collins & Frank, 2013).

This literature is far richer on intervention-level manipulations than on trait moderators: across perceptual, motor, and cognitive training, training variability robustly slows initial learning yet improves generalization, establishing task design as a primary driver of transferable learning (Raviv, Lupyan, & Green, 2022). In contrast, we know less about how stable traits and physiological regulation gate this flexibility. Stress, for example, can down-weight model-based control, with working memory buffering the effect (Otto et al., 2013). Taken together, these perspectives predict a pattern of results: training in consistent contexts promotes efficient routines, whereas switching into inconsistent contexts taxes planning/structure learning and reveals individual differences in flexibility.

R1.6. I'm not sure if this fits best within methods or discussion, but I found it surprising that the authors chose to utilize a training and transfer task with such drastically different base difficulties. Obviously it's possible to deal with that (to some degree) analytically, but

it runs counter to the approach in many domains where researchers will use training and transfer tasks that have reasonably equivalent difficulty (e.g., in perceptual research using discriminations around 45 degrees for training and 135 degrees for generalization; or in cognitive – block span for training and digit span for transfer; or in motor research one type of perturbation for training and another equivalently difficult perturbation for generalization). So it might be useful for the authors to discuss this choice either in setting up the methods or the results and how future researchers should think about that (sometimes it can't be avoided).

We agree that many domains advocate equating base difficulty across training and transfer. In our case, we intentionally allowed an asymmetry (consistent vs. inconsistent) to probe transfer across consistent-into-inconsistent and inconsistent-into-consistent. We now (1) clarify that our inconsistent version changed which visual features signaled rules (e.g., color, speed/energy) rather than the action–outcome structure, (2) note that we displayed the current mapping continuously at the bottom of the screen and updated stimuli accordingly to mitigate working-memory load and proactive interference, and (3) explicitly flag residual perceptual bottlenecks and discuss implications and alternatives (e.g., titrating difficulty) for future work.

In the third paragraph of results, we now say:

Participants were randomly assigned to one of two task versions. In the consistent version, stimulus-response (S-R) mappings—such as the meaning of a stimulus color (e.g., red = fast-moving)—remained fixed across all rounds. In the inconsistent version, these mappings changed periodically within each round, every 50 turns (e.g., red = fast-moving, then red = most energy; see Figure 1C), introducing greater uncertainty and requiring ongoing adaptation. Crucially, the core task goals, available actions, scoring rules, and control scheme were identical across versions; only the visual cues that signaled latent properties (speed/value) were subject to re-coding in the inconsistent version. Stimuli-response mappings were always presented at the bottom of the screen such that there was not an additional memory load in the inconsistent version. This manipulation was designed to examine how unpredictability influences learning, adaptability, and performance.

We've also added the following paragraph to the Limitations section of Discussion, acknowledging the potential for increased difficulty via proactive interference:

Second, our consistent and inconsistent contexts were not equated in baseline difficulty by design: the latter was intended to model volatile S-R mappings that tax monitoring and adaptation. This asymmetry helps test directional transfer but also may increase We mitigated memory load with an always-visible mapping legend, yet future studies aiming to equalize baseline difficulty could use practice-to-criterion before transfer or pre-test–based titration to match initial accuracy across contexts prior to the switch.

R1.7. Obviously the performance data through time was non-linear and so it would be necessary to include something to deal with that fact in the analysis if the goal was to do

GLM stats. However, why was the particular form chosen (round squared). That form seems to violate what we know about human learning and I'm not sure it's actually a good fit to the data (most human learning data of this sort at the individual level is best fit by something from the exponential family/exponential to asymptote). Given that so many of the inferences are drawn from the results using that particular parameterization, it would be useful to know why it was chosen (i.e., if there's theory or empirical reasons to pick it) and how the results might shift if a (potentially) better fitting form was chosen instead.
<https://pubmed.ncbi.nlm.nih.gov/10909131/>; <https://pubmed.ncbi.nlm.nih.gov/17576267/>;
<https://doi.org/10.1167/jov.21.13.5>

In the SI, we now include subject-level curve fits for three standard families: Exponential, Power, and Quadratic. We fit to each participant's acquisition data (Rounds 1–5) via nonlinear least squares. Model quality was compared per participant using BIC. Two clear patterns emerged: (1) for performance, the quadratic provided the lowest BIC distribution (power close behind; exponential typically the worst); (2) for completion time, the power function provided the lowest BIC distribution (quadratic second; exponential again worst). Crucially, all substantive inferences about Stress, Version, and their interaction were unchanged when we re-ran the mixed-effects analyses using alternative trajectory parameterizations (see SI, Fig. S1 and tables).

Our choice to report the preregistered polynomial basis (Round, Round²) in the main text was driven by both interpretability and identifiability at the group level. With five acquisition rounds, two polynomial terms flexibly capture the expected monotonic deceleration while keeping intercepts directly interpretable (performance at Round 1) and slopes interpretable as early-gain vs. deceleration effects. In contrast, a single group-level exponential basis requires a common rate parameter; when true rates are heterogeneous across individuals, the mean of exponentials is not exponential, which can mis-shape a pooled trajectory and obscure intercept/slope interpretation. Practically, with five points per person, the exponential rate is also relatively poorly identified and highly sensitive to starting values, whereas the polynomial is stable. That said, our results are robust to the functional form: whether using quadratic, power, or exponential parameterizations, the conclusions about stress, task version, and trait moderation are practically the same. We now cite and discuss these comparisons in the SI and point readers to Fig. S1, which visualizes the individual trajectories and per-participant BIC distributions.

In the Results we now say:

Following our demonstrated effect of the stress manipulation, we next aimed to explore how both stress and task characteristics together influenced task performance during the acquisition phase of the task (Rounds 1–5). We conducted two separate linear mixed effects models predicting performance scores and efficiency over the first five rounds of the learning task. Round and round squared were included as within-subjects fixed effects, and stress condition and task version were included as between-subjects fixed effects. Per-participant variations in the fixed intercept were included as a random effect. Round and round squared terms were pre-registered to investigate the polynomial trajectory of $x - x^2$. However, models investigating exponential and power trajectories of the form $e^{-\lambda x}$ and $x^{-\lambda}$ are reported in SI with near identical results. Furthermore, of these three trajectory families, the polynomial provided the best subject-level

fits to performance scores and the second-best subject-level fits to efficiency scores (second to the power function). Also see SI for models including mission type as a standalone fixed effect, in which we demonstrate that, as predicted, mission has no effect on performance nor completion time, resulting in its exclusion from models reported here.

In SI we now say:

Here we report model-family checks for the trial-to-trial learning trajectory during acquisition (Rounds 1–5). The main text mixed-effects analyses used a preregistered polynomial basis (Round, Round²) with fixed effects of Stress (MAST vs. control) and Version (consistent vs. inconsistent mappings), and a participant-specific random intercept. To verify that our inferences were not an artifact of the chosen trajectory basis, we fit subject-level curves from three commonly used families: exponential $y=ae^{\lambda x}+c$, power $y=ax^{\lambda}+c$, and quadratic $y=ax^2+bx+c$. We do this with each participant's series for two outcomes: standardized performance score and efficiency (time to complete a round, minutes). Fits used `scipy.optimize.curve_fit` with generous iteration limits (`maxfev=100000`) and participants with <5 observations were excluded. Model quality was summarized per participant by Bayesian Information Criterion $BIC=n\log(RSS/n)+k\log(n)$; lower BIC indicates better fit.

Results reproduced the mixed-effects conclusions using an entirely different fitting approach. For performance, the quadratic family yielded the lowest BIC distribution across participants (Total $BIC_{Quadratic} = -2543.66$; Mean $BIC_{Quadratic} = -12.29$), with the power model close behind and the worst fit being the exponential family (Total $BIC_{Power} = -2216.89$; Mean $BIC_{Power} = -10.71$; Total $BIC_{Exponential} = -1846.82$; Mean $BIC_{Exponential} = -8.92$). Visual inspection of fitted curves shows the expected monotonic, decelerating improvement that the quadratic basis captures well over five rounds. For completion time, the ordering shifted slightly: the power model produced the lowest BIC distribution, the quadratic was second, and exponential again under-performed (Total $BIC_{Quadratic} = -4137.80$; Mean $BIC_{Quadratic} = -19.99$; Total $BIC_{Power} = -4830.47$; Mean $BIC_{Power} = -23.34$; Total $BIC_{Exponential} = -3548.24$; Mean $BIC_{Exponential} = -17.14$). This pattern is consistent with multiplicative (percentage-like) gains in speed across rounds, versus more additive, saturating change in accuracy. Importantly, whether trajectories were parameterized as quadratic, power, or exponential did not change any qualitative inferences about Stress or Version from the mixed-effects models; the family choice mainly affected descriptive goodness-of-fit at the subject level.

Supplementary Figure 1. Top row (Performance): Left three panels show per-participant fitted curves (gray lines) for Exponential, Quadratic, and Power families; x-axis is Round (1–5), y-axis is Standardized Performance Score. Right panel shows per-participant BIC values (blue points) by family with the across-participant mean (black line). Lower BIC indicates better fit. Bottom row (Time): Same layout for Time to Complete Round (Minutes). Quadratic provides the best overall subject-level fit for performance; Power provides the best overall subject-level fit for time; Exponential is generally the worst fit for both outcomes.

R1.8. There’s a peculiar drop in performance for the inconsistent group during round 5 (one that doesn’t seem like it could be due to random chance – it’s a huge deviation from what would be expected given the trend from rounds 1-4 and the sample size). Do the authors have a reason for that? There doesn’t seem to be any change in the size of the error bars for that point, which I find surprising (I would have guessed that it was due to some

participants fatiguing or something like that, but that would tend to spread the distribution out and thus cause broader error bars).

We agree that this drop warrants further discussion. The only factor predicting this drop at round 5 is task version – people in the Inconsistent version of the task experience the drop, whereas people in the Consistent version do not. We believe that this may be associated with fatigue due to the Round 6 being unanticipated by participants, and people in the Inconsistent version (which is more taxing) preparing to finish the experiment. However, we do not have data to support this fatigue hypothesis. We now say the following in Discussion:

At the intervention level, task stability (consistent vs. inconsistent) during training emerged as a key predictor of performance. Participants trained in consistent environments with stable stimulus–response mappings demonstrated superior efficiency during the acquisition phase compared to those in inconsistent contexts. These results align with prior research emphasizing the role of task structure in promoting skill acquisition and transfer (3,25). A small, version-specific nonlinearity was also evident: in the Inconsistent condition, performance increased from Rounds 1–4, dipped at Round 5, then recovered in Round 6 (Fig. 2C). This Round-5 dip was predicted only by task version (Inconsistent) and not by switching, and we interpret it as a plausible fatigue/effort-allocation effect near the perceived end of training. This may be potentially amplified by the greater demands of the Inconsistent mapping, although we lack direct fatigue measures and therefore treat this interpretation cautiously.

R1.9. Given that the focus was on individual differences, it's a bit disappointing to not have any individual difference plots. It's honestly a little dissatisfying to only have aggregated plots when the focus is on how people differ (it would also be useful for readers to be able to see what some of the individual difference level distributions look like rather than just model outputs).

To address this point, we have added individual-trajectory visualizations to the SI. These plots overlay every participant's path colored by their continuous trait value. Our main-text figures are intended to communicate the between-person effects of continuous emotion–cognition traits on learning (i.e., how trajectories covary with trait level) rather than to profile individuals per se. The inferential model estimates trait-by-round (and Stress/Version) effects at the group level with participant-specific random intercepts; plotting hundreds of individual paths in the main text would obscure these contrasts through overplotting and make it harder to read the marginal effects that the models test. For readability, we therefore show high vs. low trait splits (mean-split displays) to visualize the direction and magnitude of moderation, consistent with the group-level parameters we report.

Figures included in SI are now:

Supplementary Figure 2. Post-switch change from Round 5 to 6 in performance (top) and completion time (bottom) by training version. Each grey line is one participant, with shade intensity proportional to a continuous trait (top left: cognitive reappraisal; top right: intolerance of uncertainty; bottom left: behavioral inhibition; bottom right: behavioral activation fun-seeking). Blue lines = average post-switch change for consistent trained, Orange lines = average post-switch change for inconsistent trained.

Reviewer 2

R2.1. Thank you for granting me the opportunity to review this manuscript. I think the manuscript has adopted an interesting experimental design and some valuable measures. Also, the data-analysis seems to have been done in a solid manner. This makes this paper potentially valuable for both research and practice.

Thank you for your thoughtful evaluation and for seeing the value in this research.

R2.2. However, I have serious concerns with regard to the theoretical foundation of the paper. The current version lacks a clear line of argumentation for the relevance of the manuscript and a theoretical lens used to conceptualize the research questions. Also, the substantiation for why specific choices were made is described only limitedly. For example, why were specific predictors chosen? Which theory or empirical findings back up these choices? And why are the employed measures a good alternative for self-reported measures? Finally, there is a lack of structure in the manuscript as specific information described in for example the results section needs to be described in the method section, which also applies for information presented in the present study section.

In general, I think a considerable improvement in structure, argumentation and theory is required to make this manuscript publishable. Below, more detailed comments per section are provided.

In this revision, we specify a clearer framework by defining transfer across domains and, per other reviewer comments, ground our rationale in the decision-making under uncertainty literature (model-based/model-free control and arbitration). We also explicitly justify each predictor: intervention-level volatility and stress (as primary task determinants of generalization), person-level emotion-cognition traits (emotion regulation, intolerance of uncertainty, and behavioral inhibition and activation system sensitivity) as moderators of flexibility, and physiological indices (heart rate, blood pressure, heart rate variability, and sympathetic response indicators) as objective markers of reactivity/regulation that complement and sometimes outperform self-report.

We initially had the Methods section at the end to align with Nature-style formatting, but have realized that Communications Psychology adheres to the more traditional Methods-Results-Discussion format. We have restructured the manuscript accordingly.

R2.3. The theoretical framework is brief at its current form. It lacks a clear theoretical lens based on which a research rationale has been worked out. Also different concepts could be explained more thoroughly (i.e., definitions of concepts) and how they relate to one another. I also miss a clear definition of transfer. What is your definition of transfer? And how is this operationalized?

We have expanded the theoretical lens and clarified definitions. The Introduction now (1) distinguishes domains of transfer (episodic/context-dependent memory; perceptual/motor learning; educational/cognitive training; cognitive control/rule learning), (2) gives a clear definition of transfer for this paper, and (3) situates our rationale in the decision-making-under-uncertainty framework (model-based/model-free control and arbitration).

In the Introduction, we now say the following, which includes a domain-general definition for transfer:

One reason for the lack of agreement is that researchers are inconsistent in how context is operationalized when examining near and far transfer (Barnett & Ceci, 2002). Across domains,

context denotes different things. In episodic memory, context often means the physical environment (e.g., underwater vs. on land; context-dependent memory) (Godden & Baddeley, 1975; Murre, 2021; Smith & Vela, 2001). In motor and perceptual skill learning, context typically refers to sensorimotor/task constraints and training settings (e.g., simulation-based surgical skills, sport perceptual training) (Dawe et al., 2014; Hadlow et al., 2018). In educational/cognitive training, context frequently refers to the similarity of stimuli or underlying cognitive operations (near vs. far transfer) (Kassai et al., 2019; Melby-Lervag & Hulme, 2013; Sala & Gobet, 2019; Shipstead et al., 2012; Simons et al., 2016; Spearman, 1904; Thurstone, 1938; Pan & Rickard, 2018). In cognitive control/rule learning, context can denote stimulus–response or rule mappings that vary over time (variably mapped/dynamic tasks) (Macnamara & Frank, 2018; Wen et al., 2023). Across these literatures, ‘transfer’ refers to the extent to which performance and efficiency acquired in a trained context are maintained or appropriately adapted when contextual features change. More recently, a computational and neural framework has been proposed to unify generalization across such contexts (Heald et al., 2023), alongside classic individual-difference perspectives on skill learning (Ackerman, 1987).

We also include the following paragraphs anchoring our choice in the decision-making under uncertainty literature:

A complementary tradition in decision-making under uncertainty distinguishes between model-based (i.e., planning using an internal model or ‘cognitive map’) and model-free (i.e., habitual, cached-value) control, with critical implications for transfer. Model-based control supports flexible generalization when contingencies change, whereas model-free control yields efficient performance in stable contexts but poorer transfer under revaluation or structural shifts (e.g., goal or transition changes; Daw et al., 2011; Doll, Simon, & Daw, 2012; Behrens et al., 2018). A successor representation stores a predictive map of which situations are likely to come next; it enables partial transfer when goals (rewards) change but the underlying transition structure is preserved (Gershman et al., 2023; Momennejad et al., 2017). People shift between planning and habit when they judge that the benefits of being flexible outweigh the mental effort, and when the task makes planning feasible (Kool, Cushman, & Gershman, 2017; Collins & Frank, 2013).

The literature is far richer on intervention-level manipulations than on trait-level predictors: across perceptual, motor, and cognitive training, training variability robustly slows initial learning yet improves generalization, establishing task design as a primary driver of transferable learning (Raviv, Lupyan, & Green, 2022). In contrast, we know less about how stable traits and physiological regulation influence this flexibility. What we know about individual traits tend to come from their interaction with intervention-level effects. Stress, for example, can down-weight model-based control, with working memory buffering the effect (Otto et al., 2013). Taken together, intervention- and trait-level interactions predict a pattern of results: training in consistent contexts promotes efficient routines, whereas switching into inconsistent contexts taxes planning/structure learning and reveals individual differences in flexibility.

R2.4. At the beginning of the introduction you state that there is disagreement among researchers in what constitutes transfer and which predictors should be included. Based on the current version, it is unclear how the approach suggested by you will be a step forward in this discussion, nor how your study would be a contribution for practitioners and/or trainers. This makes it difficult for me as a reader to understand what the utility is of your study.

We now more clearly how our approach advances the transfer debate and why it matters for practitioners. We frame our contribution as a multi-level account that (1) manipulates task structure (consistent vs. inconsistent mappings) and acute stress (intervention-level), (2) measures emotion-cognition traits and physiological regulation (between-person), and (3) models performance changes with practice (within-person). We then show how these layers jointly shape transfer. Theoretically, we connect consistent vs. inconsistent contexts to model-free vs. model-based control, which predict asymmetric transfer and trait-contingent adaptation. Practically, we translate these into possible recommendations for improved transfer. We now say this at the end of the Introduction section:

This study advances transfer theory by jointly manipulating task structure and stress while modeling between-person traits and within-person practice. Consistent contexts should foster efficient, model-free routines that transfer poorly to volatile settings; inconsistent contexts should recruit model-based planning (and SR-like predictive maps), supporting transfer when contingencies change but taxing learners unevenly depending on stress reactivity and emotion-cognition traits. For practitioners, this multi-level account yields clear factors for training variability, stress context intervention, and trait-informed scaffolds, all to optimize transfer rather than assume it.

R2.5. Even though I acknowledge your approach towards pre-registering your hypotheses, it is still very important for the reader to understand why you expect specific findings. Therefore, I would recommend working out a rationale for why you formulated specific hypotheses and to back up those claims by means of different empirical findings and/or theory.

We have now added the following paragraphs to the introduction to better establish the decision-making under uncertainty literature and what it predicts of individual differences in transfer:

A complementary tradition in decision-making under uncertainty distinguishes between model-based (i.e., planning using an internal model or ‘cognitive map’) and model-free (i.e., habitual, cached-value) control, with critical implications for transfer. Model-based control supports flexible generalization when contingencies change, whereas model-free control yields efficient performance in stable contexts but poorer transfer under revaluation or structural shifts (e.g., goal or transition changes; Daw et al., 2011; Doll, Simon, & Daw, 2012; Behrens et al., 2018). A successor representation stores a predictive map of which situations are likely to come next; it enables partial transfer when goals (rewards) change but the underlying transition structure is preserved (Gershman et al., 2023; Momennejad et al., 2017). People shift between planning and

habit when they judge that the benefits of being flexible outweigh the mental effort, and when the task makes planning feasible (Kool, Cushman, & Gershman, 2017; Collins & Frank, 2013).

This literature is far richer on intervention-level manipulations than on trait-level predictors: across perceptual, motor, and cognitive training, training variability robustly slows initial learning yet improves generalization, establishing task design as a primary driver of transferable learning (Raviv, Lupyan, & Green, 2022). In contrast, we know less about how stable traits and physiological regulation moderate this flexibility. One exception is stress, which can down-weight model-based control, with working memory buffering the effect (Otto et al., 2013). Taken together, these intervention- and trait-level interactions predict a pattern of results for transfer: training in consistent contexts promotes efficient routines, whereas switching into inconsistent contexts taxes planning/structure learning and reveals individual differences in flexibility. However, it is unknown how emotion-cognition traits, which interact with stress and environment, predict learning transfer in different contexts.

R2.6. Specific elements described in the present study would fit better in a method section. Especially regarding the sample and measurement instruments.

We have removed elements specific to the sample and measurement instruments from the Present Study, and now introduce them in the beginning of the Results section (where pertinent for understanding the results). We also elaborate on sample and measurement in the Methods section.

R2.7. It is potentially interesting that you use more objective measures such as blood pressure and electrocardiograms. This is a step forward in comparison to self-reported measures. Nevertheless, choices for specific types of measures should be justified more clearly with theory. Information about its validity and previous studies using these methods is important as this can help the reader understand why your approach is valid and reliable. This also applies for the intervention manipulations you mention in the present study section. It is unclear why these manipulations were made and whether they were used in previous studies.

We have strengthened the theoretical justification for psychophysiological measures and the stressor along three points. First, we clarify why objective physiology reduces demand characteristics and improves response-process validity relative to self-report. Second, we note the validity of heart rate variability for indexing vagal control. Third, we explain how the stress manipulation robustly activates the sympathetic nervous system and often the hypothalamic–pituitary–adrenal axis, with stronger cortisol when social evaluation is added; we measured autonomic responses and note specific limitations. In the Methods we now say:

Cold-pressor–based stressors, including the MAST, robustly engage both the autonomic nervous system and the hypothalamic–pituitary–adrenal (HPA) axis, with numerous studies reporting elevations in salivary cortisol following these tasks (e.g., Liu et al., 2024; Minkley et al., 2014;

Schwabe et al., 2008; Schwabe & Schächinger, 2018). In the present study, we measured autonomic indices of stress reactivity: heart rate and mean arterial pressure (primarily reflecting sympathetic activation) and heart-rate variability metrics—RMSSD and high-frequency power—as markers of parasympathetic/vagal activity, complemented by CSI and low-frequency power as sympathetic proxies and CVI as a parasympathetic proxy (e.g., Appelhans & Luecken, 2006; Toichi et al., 1997; Pham et al., 2021).

R2.8. I would recommend to first describe your methodology and then to move on towards the results and discussion section. This would also increase the likelihood that the right information is presented in the right section. I now read information related to the methods in the results section (see first alinea until at least manipulation checks) and in the present study section. In general it would make it easier for the reader to understand why you made specific choices.

We appreciate the difficulty in reading with this structure. We initially had the Methods section at the end to align with Nature-style formatting, but have realized that Communications Psychology adheres to the more traditional Methods-Results-Discussion format. We have restructured the manuscript accordingly.

R2.9. A context description of your study would be of help to understand why a game setting has been used, and how this relates to real life tasks of your sample (or not). This is especially important as you also mention this as a limitation in your discussion section.

We have added information in the Methods and briefly in the Introduction: Our grid-world is a microworld for dynamic decision-making. There are interdependent choices, shifting contingencies, and resource allocation all of which are commonly used to study transfer and individual differences. Simulation/game training shows transfer to operational settings in several domains (e.g., surgery), though effect sizes vary with design fidelity and variability. In the Methods we now say:

We administered a six-round learning task that required participants to manage structures on a grid-world, gather resources, and defend against invaders¹⁷. Such simulated environments can support skill transfer when they capture structural features of target tasks (e.g., variability, feedback, time-pressure).

R2.10. Information about the reliability and validity of the used questionnaires should be presented in the method section and not in the result section.

We now (1) state instrument reliability (α/ω) in the Materials subsection of Methods.

R2.11. The procedure section seems to be worked out comprehensively. Although I must also admit that I am not an expert on the used procedure. Perhaps other reviewers provide valuable feedback on this.

Thank you for the positive feedback. We are pleased to hear that the procedure reads clearly.

R2.12. I would recommend describing a data-analysis section in which you explain your statistical approach towards analyzing the data.

We have added an analysis map table linking research questions/hypotheses to models. The added text in Results reads as follows:

We preregistered primary models and report them in the order below. Table 2 maps each research question to the corresponding model, predictors, interactions, and outcomes.

Table 2.

Analysis map of hypotheses (H1–H4) to models, predictors, interactions, and outcomes

Hypothesis	Phase / Dataset	Model	Key Predictors (fixed)	Key Interactions	Outcome(s)	Prereg.
H1: Consistent > Inconsistent during acquisition (performance, efficiency)	Acquisition (Rounds 1–5)	LME: $y \sim$ Round + Round ² + Version + Stress + (1 id)	Round, Round ² ; Version (Certain vs Uncertain); Stress (MAST vs Control)	Round×Version; Round×Stress; Version×Stress	Standardized Performance; Completion Time (min)	Yes
H2: Lower stress > Higher stress (performance, efficiency)	Acquisition (Rounds 1–5)	Same LME as H1 (reported jointly)	As above	As above	Standardized Performance; Completion Time	Yes
H3: Practice interacts with training & post-training change to predict transfer	Transfer (Round 6 vs Round 5)	LME: $y \sim$ Phase (R5/R6) + Training + Version + Switch (Yes/No) + Phase×Version + Phase×Switch + Version×Switch + (1 id)	Phase (pre/post); Version; Switch	Phase×Version; Phase×Switch; Version×Switch; Phase×Version×Switch	ΔPerformance (R6–R5); ΔTime (R6–R5) or Level at R6 with R5 covariate	Yes
H4: Emotion–cognition traits moderate acquisition & transfer	Acquisition (R1–R5) and Transfer (R5→R6)	LME (Acq): $y \sim$ Round + Round ² + Version + Stress + Trait + 2-way/3-way interactions + (1 id) LME (Trans): $\Delta y \sim$ Version + Switch + Trait	Trait (e.g., ERQ, reappraisal, IUS-12, BIS/BAS); Version; Stress; Round/Phase	Trait×Version; Trait×Stress; Trait×Round (or Trait×Phase); selected 3-ways (e.g., Trait×Version×Phase)	Standardized Performance; Completion Time	Yes

Note. LME = Linear Mixed-Effects model; Version = training environment (Consistent vs Inconsistent); Stress = MAST vs Control; Trait = emotion–cognition trait (ERQ reappraisal, IUS-12, BIS/BAS); Phase = pre/post switch (Round 5 vs 6).

R2.13. The result section provides a lot of information and details about the analyses done to investigate your research aims. For the reader, it would help if you could describe how different statistical analyses serve the research purposes. Why do you perform specific analyses? And which research questions are answered by doing this? In the current version there are no clear research questions, whereas this would support you in working out a comprehensive structure in your results section. Another option is to do so based on your hypotheses, to follow that structure in your result section and to briefly mention each hypothesis (or several) per analysis.

We have added the table mapping the analyses to the hypotheses (see response to 12 above). We have also included tags for H1-H4 in the Present Study section of Introduction, and refer to H1-H4 throughout results to indicate where evidence supports or fails to support these hypotheses. The Present Study section now reads as follows:

We preregistered several hypotheses, separately for anticipated effects of learning during training (https://osf.io/wj68f/?view_only=2dbf80a1928c411eb7c79daf990269e0), and transfer post-training (https://osf.io/x72z3/?view_only=da1984f9ff244ca798fa6359e1b749b7). At the intervention level, we hypothesized that performance would be greater in the consistent condition than the inconsistent condition (H1), and that performance would be greater in a less stressful context than a more stressful context (H2). We further hypothesized that changes in amount of practice would interact with training condition (consistent vs. inconsistent) and post-training condition change (e.g., switching from consistent to inconsistent) to predict learning and transfer (H3). Finally, we hypothesized that several emotion-cognition traits at the between-person (i.e., individual differences) level would interact with training condition and stress context at the intervention level, including elements of emotion regulation, intolerance of uncertainty, and behavioral inhibition (H4). Through a series of models, we demonstrate the contributions of each level of analysis (i.e., intervention, between-person, within-person) in understanding individual differences in acquisition and transfer of learned skills, and how those differences deviate from group-level findings.

Each section of Results now calls to these hypotheses H1-H4 to clearly indicate which analyses were used to test which hypotheses, and whether the results of those analyses support the hypotheses.

R2.14. In the theoretical framework you mention that you look at both within- and between-level effects. However, this is not mentioned again at the beginning of the

discussion section as you only mention the within-level there. Does this imply that you have not looked at the between-person level? In any case, consistency is warranted here.

We now open the Discussion by recapping both levels to maintain consistency with the framework: (1) within-person practice effects (decelerating curves; efficient adaptation of completion time after switching), and (2) between-person moderation by traits and physiology (e.g., reappraisal, intolerance of uncertainty, HRV), clarifying that both were tested and reported. The opening of the Discussion section now reads:

Our findings integrate within-person practice dynamics with between-person trait/physiology moderators to explain why group means can mask compensatory adaptations at transfer.

R2.15. I also miss linkages with the formulated hypotheses. To what extent were your findings in line with your expectations and for which hypotheses did you not find support? How does it relate to previous studies? And how could unexpected findings be explained?

We have included tags for H1-H4 in the Present Study section of Introduction, and refer to H1-H4 throughout results to indicate where evidence supports or fails to support these hypotheses. The Present Study section now reads as follows:

We preregistered several hypotheses, separately for anticipated effects of learning during training (https://osf.io/wj68f/?view_only=2dbf80a1928c411eb7c79daf990269e0), and transfer post-training (https://osf.io/x72z3/?view_only=da1984f9ff244ca798fa6359e1b749b7). At the intervention level, we hypothesized that performance would be greater in the consistent condition than the inconsistent condition (H1), and that performance would be greater in a less stressful context than a more stressful context (H2). We further hypothesized that changes in amount of practice would interact with training condition (consistent vs. inconsistent) and post-training condition change (e.g., switching from consistent to inconsistent) to predict learning and transfer (H3). Finally, we hypothesized that several emotion-cognition traits at the between-person (i.e., individual differences) level would interact with training condition and stress context at the intervention level, including elements of emotion regulation, intolerance of uncertainty, and behavioral inhibition (H4). Through a series of models, we demonstrate the contributions of each level of analysis (i.e., intervention, between-person, within-person) in understanding individual differences in acquisition and transfer of learned skills, and how those differences deviate from group-level findings.

Each section of Results now calls to these hypotheses H1-H4 to clearly indicate which analyses were used to test which hypotheses, and whether the results of those analyses support the hypotheses.

R2.16. I would recommend working out a separate paragraph in which you describe your theoretical implications of your study. You now do describe some contributions of your

study in the last alinea of the discussion, but this is still relatively limited. This would support the rationale for why your study has value.

We have added a dedicated Theoretical Implications subsection in the Discussion section with the following:

The asymmetries we observe, including performance disruption when moving from consistent to inconsistent contexts, alongside rapid convergence in efficiency, are consistent with a dual-systems account in which model-free routines optimize stable contexts, while model-based control supports reconfiguration under volatility at a cost. Our trait and physiology results suggest that emotion regulation, uncertainty tolerance, and HRV change moderate flexibility between these two strategies. Integrating these person-level moderators with intervention-level determinants helps explain why group means can mask compensatory adaptations at transfer and yields concrete predictions: transfer success depends jointly on whether change targets rules vs. goals, on practice history, and on EC profiles that govern willingness and capacity to re-plan. By formalizing these interactions, the present work extends structural accounts of transfer to a multi-level framework in which environment design, internal control architecture, and emotion-cognition dynamics jointly determine when and how skills generalize.

R2.17. In addition to that, an additional paragraph in which you describe the practical implications of your study in a more concrete manner would be of value as well. In the current version, the practical implications are only described superficially which hampers the value of your study for practice.

We have now added a dedicated Implications subsection to the Discussion that first addresses practical implications, then theoretical implications. In Implications, we describe how our findings can inform education, workforce training, and personalized learning through a two-armed approach that adjusts both training environments (e.g., structured variability, staged switches) and support structures tailored to emotion-cognition (EC) traits (e.g., reappraisal coaching, scaffolds for uncertainty). Also, in Implications, we describe how incorporating EC traits and physiological regulation into model-based/model-free accounts clarify when transfer succeeds or falters. The new Implications subsection in the Discussion now reads as follows:

Our results suggest a two-armed strategy for designing training that transfers: tune the environment and tailor to the person. On the environment side, introducing structured variability (e.g., controlled re-mapping of stimulus–response rules, deliberate mid-course switches) may slow early learning but prepare learners for changes at deployment. On the person side, brief assessments of emotion-cognition traits (e.g., intolerance of uncertainty, cognitive reappraisal) and feasible physiological indices of adaptive regulation (e.g., HRV change where instrumentation is available) may inform adaptive scaffolding, such as providing advance cues and rule legends for high-uncertainty learners, reappraisal coaching for learners low in cognitive reappraisal, and extra practice in volatile contexts for those who show limited adaptive regulation. Future research can explore whether these findings extend to educational and workforce settings to predict person–job fit and personalized learning. For example, placing

learners whose emotion-cognition profiles favor routine stability into roles or modules emphasizing consistent mappings, or using variability-rich practice to cultivate flexibility in those slated for volatile environments.

The asymmetries we observe, including performance disruption when moving from consistent to inconsistent contexts, alongside rapid convergence in efficiency, are consistent with a dual-systems account in which model-free routines optimize stable contexts, while model-based control supports reconfiguration under volatility at a cost. Our trait and physiology results suggest that emotion regulation, uncertainty tolerance, and heart rate variability change moderate flexibility between these two strategies. Integrating intervention-level determinants with these person-level moderators helps explain why group means can mask compensatory adaptations at transfer and yields concrete predictions. That is, transfer success depends jointly on whether change targets rules vs. goals, on practice history, and on trait profiles that govern willingness and capacity to re-plan. By formalizing these interactions, the present work extends structural accounts of transfer to a multi-level framework in which environment design, internal control architecture, and emotion-cognition dynamics jointly determine when and how skills generalize.

Reviewer 3

R3.1. I want to sincerely thank the authors for their valuable work. The authors present a meaningful and timely investigation into the role of individual emotion-cognition traits in learning transfer under stress, which a topic with clear relevance for both psychological theory and practical training applications. The use of context switching and stress manipulation in a learning task, combined with physiological and self-report measures, offers a valuable contribution to the growing body of work on how affective and cognitive systems interact during complex learning processes. I personally think the manuscript would benefit from several important revisions to improve clarity, structure, and theoretical framing. The literature review could more clearly articulate the empirical and theoretical gaps the study addresses. Additionally, key design elements—such as task environments, conditions, and individual difference measures—need more background and justification. And the Discussion would be strengthened by more explicitly connecting the findings to theoretical frameworks and clarifying how this work advances the field.

Thank you for the thoughtful summary and guidance. In this revision we have (1) added an explicit gap/contribution paragraph to the end of the Introduction; (2) expanded background and justification for task environments (consistent vs. inconsistent) and the stress manipulation, clarifying physiological/self-report measures and adding HPA-axis citations; (3) strengthened the Discussion by linking findings to model-based/model-free and uncertainty frameworks and adding a dedicated Implications subsection (practical vs. theoretical); and (4) improved clarity/structure by labeling hypotheses H1–H4 and referencing support in Results, with clearer pointers from Results to Methods. We believe these changes address the concerns raised.

R3.2. The structure and organization of the manuscript, particularly the Methods and Results sections, require revision for clarity and coherence. First, I am not sure why the Method section is the last main section of the manuscript (after the Discussion section). Is it an accident? It is quite confusing to read without knowing the detail of the measures and tasks. Second, a substantial portion of content currently placed at the beginning of the Results section (up to the “Manipulation Check” heading) describes the task and experimental procedures and would be more appropriately located in the Methods section. To improve readability and align with standard reporting practices, I recommend reorganizing the manuscript so that all task descriptions, measures, and procedures are clearly presented within the Methods section, followed by the Results.

We initially had the Methods section at the end to align with Nature-style formatting, but have realized that Communications Psychology adheres to the more traditional Methods-Results-Discussion format. We have restructured the manuscript accordingly.

R3.3. You mentioned that “physiological stress responses were measured using electrocardiogram, heart rate, and blood pressure”. While these are valuable indicators of autonomic activity, particularly sympathetic nervous system (SNS) activation, it is important to note that cold pressor tasks also robustly engage the hypothalamic-pituitary-adrenal (HPA) axis. Prior research has frequently included salivary cortisol (as you noted in the limitation section) as a key biomarker of HPA axis activation and a reliable measure of stress response in addition to blood pressure. (For example: Liu et al., 2024; Minkley et al., 2014; Schwabe et al., 2008, Schwabe & Schächinger, 2018, etc.) Including relevant citations would help justify the use of these measures (i.e., blood pressure, HRV, heart rate) and clarify how they reflect stress reactivity.

We have revised the Methods to explicitly note HPA involvement, added citations documenting cortisol responses to cold pressor/MAST (e.g., Liu et al., 2024; Minkley et al., 2014; Schwabe et al., 2008; Schwabe & Schächinger, 2018), and clarified our rationale for focusing on autonomic measures (HR, MAP, HRV; with CSI/LF and CVI as sympathetic/parasympathetic proxies):

Cold-pressor-based stressors, including the MAST, robustly engage both the autonomic nervous system and the hypothalamic–pituitary–adrenal axis, with numerous studies reporting elevations in salivary cortisol following these tasks (e.g., Liu et al., 2024; Minkley et al., 2014; Schwabe et al., 2008; Schwabe & Schächinger, 2018). In the present study, we measured autonomic indices of stress reactivity: heart rate and mean arterial pressure (primarily reflecting sympathetic activation) and heart-rate variability metrics—RMSSD and high-frequency power—as markers of parasympathetic/vagal activity, complemented by CSI and low-frequency power as sympathetic proxies and CVI as a parasympathetic proxy (e.g., Appelhans & Luecken, 2006; Toichi et al., 1997; Pham et al., 2021).

We also updated the Limitations to state explicitly that we did not assay cortisol and therefore cannot directly assess HPA responses:

Moreover, while our physiological measures (e.g., heart rate variability, blood pressure) provide meaningful insight into autonomic reactivity, future research incorporating additional biomarkers—such as cortisol, impedance cardiography, or galvanic skin response—could yield a more comprehensive profile of stress-related changes. Although the MAST reliably recruits the HPA axis, we did not assay HPA-axis hormones (e.g., salivary cortisol) and therefore cannot directly characterize endocrine stress responses in this study.

R3.4. The introduction provides a broad overview of learning transfer and individual differences, but it would benefit from a more clearly articulated theoretical or empirical gap. While the literature review outlines relevant prior work, it does not sufficiently specify what existing studies have found, where they fall short, or how this manuscript addresses those limitations. The authors could explicitly highlight more on what is currently missing in the literature and clearly state how their study contributes novel insights or methodological advancements beyond prior research.

We have added the following paragraph near the end of the Introduction (immediately before the preregistered hypotheses), which (1) contrasts the comparatively rich intervention-level literature with the sparser trait/physiology literature, (2) specifies what prior work shows and where it falls short, and (3) states the resulting predictions our study was positioned to test:

The literature is far richer on intervention-level manipulations than on trait-level predictors: across perceptual, motor, and cognitive training, training variability robustly slows initial learning yet improves generalization, establishing task design as a primary driver of transferable learning (Raviv, Lupyan, & Green, 2022). In contrast, we know less about how stable traits and physiological regulation gate this flexibility. What we know about individual traits tend to come from their interaction with intervention-level effects. Stress, for example, can down-weight model-based control, with working memory buffering the effect (Otto et al., 2013). Taken together, intervention- and trait-level interactions predict a pattern of results: training in consistent contexts promotes efficient routines, whereas switching into inconsistent contexts taxes planning/structure learning and reveals individual differences in flexibility.

R3.5. Is there a specific reason that you only recruited participants from 18 - 30 years old? Age is a well-established factor influencing cognitive abilities, emotional regulation, and learning strategies—all of which are central to the current study. Limiting the sample to younger adults may restrict the generalizability of the findings, particularly given that older and middle-aged adults often differ meaningfully in the psychological constructs under investigation.

We limited enrollment to 18–30 primarily to minimize age-related heterogeneity in autonomic physiology (e.g., baseline HRV and reactivity, blood pressure) that could obscure stress-

reactivity and transfer estimates, and to align with our half-university undergraduate sample. We now state this rationale explicitly in Methods:

“Two-hundred fifty-six healthy adults participated in the experiment. Half the participants were recruited from the university’s SONA subject pool, the other half from the local Cleveland community, in exchange for course credit or \$80 USD, respectively. All participants were required to be between 18 and 30 years of age, proficient in the English language, and had no self-reported history of heart disease, peripheral vascular disease, venous thrombosis, diabetes mellitus, fainting/loss of consciousness, seizures, frostbite, loss of feeling in limbs, or Reynaud’s syndrome. Cardiology researchers have reported that measures such as heart-rate variability change with age (Umetani et al., 1998; Bonnemeier et al., 2003), so we restricted the sample to 18–30 years of age to minimize age-related heterogeneity in autonomic physiology that could confound stress-reactivity and transfer estimates in our sample.”

We also acknowledge the narrowed generalizability in Limitations, noting that future work should examine age as a moderator:

“Additionally, while the task incorporated elements of stress and uncertainty through contextual manipulations, it did not target broader or more varied skill domains, such as multitasking or strategic planning in open-ended environments. Moreover, although the task required some degree of dynamic adaptation as it unfolded over time and conditions changed for some participants, it does not capture the full range of adaptive demands present in real-world learning contexts. Finally, our sampling frame was restricted to younger adults (18–30), which limits generalizability to middle-aged and older adults; future work could explicitly test age as a moderator of stress reactivity and transfer.”

R3.6. A more detailed explanation of key experimental variables including “Consistent vs. Inconsistent” task environments and the “Stress vs. Non-stress” conditions in the introduction section. These factors appear to play a central role in the study design and theoretical framework, yet their relevance is not fully explained. The authors should elaborate on why these variables were selected, how they relate to learning transfer, and what prior research suggests about their potential effects.

In line with this and similar comments by other reviewers, we have revised the Introduction to situate our ‘uncertainty/variably mapped’ framing within the decision-making under uncertainty literature and how this connects with learning transfer:

A complementary tradition in decision-making under uncertainty distinguishes between model-based (i.e., planning using an internal model or ‘cognitive map’) and model-free (i.e., habitual, cached-value) control, with critical implications for transfer. Model-based control supports flexible generalization when contingencies change, whereas model-free control yields efficient performance in stable contexts but poorer transfer under revaluation or structural shifts (e.g., goal or transition changes; Daw et al., 2011; Doll, Simon, & Daw, 2012; Behrens et al., 2018). A successor representation stores a predictive map of which situations are likely to come next; it

enables partial transfer when goals (rewards) change but the underlying transition structure is preserved (Gershman et al., 2023; Momennejad et al., 2017). People shift between planning and habit when they judge that the benefits of being flexible outweigh the mental effort, and when the task makes planning feasible (Kool, Cushman, & Gershman, 2017; Collins & Frank, 2013).

This literature is far richer on intervention-level manipulations than on trait moderators: across perceptual, motor, and cognitive training, training variability robustly slows initial learning yet improves generalization, establishing task design as a primary driver of transferable learning (Raviv, Lupyán, & Green, 2022). In contrast, we know less about how stable traits and physiological regulation influence this flexibility. Stress, for example, can down-weight model-based control, with working memory buffering the effect (Otto et al., 2013). Taken together, these perspectives predict a pattern of results: training in consistent contexts promotes efficient routines, whereas switching into inconsistent contexts taxes planning/structure learning and reveals individual differences in flexibility.

R3.7. Did you conduct a power analysis to determine the sample size for each condition?

In our pre-registration, we note the following regarding our sample size justification:

252 participants is based on several rationales. In our previous research using the same learning task and a similar design, we collected data and found significant effects with 128 participants. In the current study, we are also examining individual differences, a sample size of 240 allows us to have over 60 participants in each between-subjects condition to investigate associations, a common rule of thumb in correlational research. Finally, to mitigate potential data loss, we will collect data from 252 participants.

We now make this explicit in the Methods section.

R3.8. I might have missed something here “Round and round squared were included as within-subjects fixed effects...” Why round 2 was included as predictor in the model?

Round squared was included in the model as a within-subjects fixed effect to capture curvilinear changes in performance. Together with the effect of Round, the Round - Round² polynomial would describe a decelerating learning curve.

R3.9. In Figure 2C, I noticed an interesting performance trajectory in the inconsistent condition: performance increases steadily from Round 1 to Round 4, then drops noticeably in Round 5, before recovering again in Round 6. This non-linear pattern, especially the decline in Round 5, stands out and may warrant further discussion. Could you further clarify or provide explanation to this? Does this dip corresponds with a specific task manipulation (e.g., switching context) or participant-level factors (e.g., fatigue, adjustment period)?

We agree that this drop warrants further discussion. The only factor predicting this drop at round 5 is task version – people in the Inconsistent version of the task experience the drop, whereas people in the Consistent version do not. We believe that this may be associated with fatigue due to the Round 6 being unanticipated by participants, and people in the Inconsistent version (which is more taxing) preparing to finish the experiment. However, we do not have data to support this fatigue hypothesis. We now say the following in the Discussion:

At the intervention level, task stability (consistent vs. inconsistent) during training emerged as a key predictor of performance. Participants trained in consistent environments with stable stimulus–response mappings demonstrated superior efficiency during the acquisition phase compared to those in inconsistent contexts. These results align with prior research emphasizing the role of task structure in promoting skill acquisition and transfer (3,25). A small, version-specific nonlinearity was also evident: in the Inconsistent condition, performance increased from Rounds 1–4, dipped at Round 5, then recovered in Round 6 (Fig. 2C). This Round-5 dip was predicted only by task version (Inconsistent) and not by switching, and we interpret it as a plausible fatigue/effort-allocation effect near the perceived end of training. This may be potentially amplified by the greater demands of the Inconsistent mapping, although we lack direct fatigue measures and therefore treat this interpretation cautiously.

R3.10. You mentioned that in the discussion “cognitive reappraisal, intolerance of uncertainty, and behavioral inhibition influenced how participants responded to a change in context from their training environments.” However, the effect of behavioral inhibition was not discussed.

To address this, we have revised the following paragraph in the Discussion:

Individual differences in emotion-cognition traits played a pivotal role in shaping both pre-switch performance and post-switch transfer. Notably, cognitive reappraisal, intolerance of uncertainty, and behavioral inhibition influenced how participants responded to a change in context from their training environments. For example, low cognitive reappraisers excelled in predictable consistent contexts but struggled to maintain performance scores when transitioning to inconsistent environments, potentially reflecting a reliance on task-specific strategies that did not generalize well. In their case, a reduced ability to engage in cognitive reappraisal may have limited their flexibility, favoring accuracy at the cost of adaptability and speed when faced with novel demands. Behavioral inhibition likewise shaped transfer: participants lower in behavioral inhibition showed greater divergence in efficiency after the switch, whereas those higher in behavioral inhibition exhibited more conservative speed adjustments, consistent with avoidance sensitivity constraining rapid reconfiguration.

R3.11. The manuscript would benefit from a more structured and explicit discussion of both the practical and theoretical implications of the findings. For example, the last paragraph under the “Future Direction” section (“Our findings suggest the importance of

designing training interventions...”) talks about the implication of this work, including adaptive training, emotion regulation coaching, and personalized scaffolding. I recommend adding a dedicated “Implications” subsection within the Discussion that distinguishes: (1) Practical implications, especially how the findings may inform education, workforce training, or personalized learning interventions based on emotion-cognition traits; and (2) Theoretical implications, specifically how this study advances existing models of learning transfer by incorporating emotion-cognition traits (e.g., cognitive reappraisal, intolerance of uncertainty), and how it addresses empirical or conceptual gaps in the literature. A clearer articulation of the contribution to the literature would enhance the theoretical significance.

Thank you for this suggestion. We have added a dedicated Implications subsection to the Discussion that explicitly separates (1) practical implications, such as how our findings can inform education, workforce training, and personalized learning through a two-armed approach that adjusts both training environments (e.g., structured variability, staged switches) and supports tailored to emotion-cognition traits (e.g., reappraisal coaching, scaffolds for uncertainty), and (2) theoretical implications, such as how incorporating emotion-cognition traits and physiological regulation into model-based/model-free accounts clarifies when transfer succeeds or falters. The new Implications subsection in the Discussion now reads as follows:

Our results suggest a two-armed strategy for designing training that transfers: tune the environment and tailor to the person. On the environment side, introducing structured variability (e.g., controlled re-mapping of stimulus–response rules, deliberate mid-course switches) may slow early learning but prepare learners for changes at deployment. On the person side, brief assessments of emotion-cognition traits (e.g., intolerance of uncertainty, cognitive reappraisal) and feasible physiological indices of adaptive regulation (e.g., HRV change where instrumentation is available) may inform adaptive scaffolding, such as providing advance cues and rule legends for high-uncertainty learners, reappraisal coaching for learners low in cognitive reappraisal, and extra practice in volatile contexts for those who show limited adaptive regulation. Future research can explore whether these findings extend to educational and workforce settings to predict person–job fit and personalized learning. For example, placing learners whose emotion-cognition profiles favor routine stability into roles or modules emphasizing consistent mappings, or using variability-rich practice to cultivate flexibility in those slated for volatile environments.

The asymmetries we observe, including performance disruption when moving from consistent to inconsistent contexts, alongside rapid convergence in efficiency, are consistent with a dual-systems account in which model-free routines optimize stable contexts, while model-based (and successor-representation-like) control supports reconfiguration under volatility at a cost. Our trait and physiology results suggest that emotion regulation, uncertainty tolerance, and heart rate variability change moderate flexibility between these two strategies. Integrating intervention-level determinants with these person-level moderators helps explain why group means can mask compensatory adaptations at transfer and yields concrete predictions. That is, transfer success depends jointly on whether the change requires remapping stimulus–response

rules or redefining the task objective, on practice history, and on trait profiles that govern willingness and capacity to re-plan. By formalizing these interactions, the present work extends structural accounts of transfer to a multi-level framework in which environment design, internal control architecture, and emotion-cognition dynamics determine when and how skills generalize.

R3.12. The introduction section did not have heading “Introduction”.

We have added this heading.

R3.13. In page 4, authors stated that “For example, in a probabilistic learning task, highly anxious individuals may exhibit heightened threat anticipation and overestimate the likelihood of negative outcomes, leading them to rely on rigid, well-learned strategies rather than adapting flexibly to new contingencies.” Is there any scientific evidences supporting this? (Need references)

We see how this wasn’t clear. In the sentence prior we had cited Browning et al., 2015, and had intended for the highlighted sentence to be a continuation of that. We now also cite Browning et al., 2015 in the highlighted sentence.

For example, in a probabilistic learning task, highly anxious individuals may exhibit heightened threat anticipation and overestimate the likelihood of negative outcomes, leading them to rely on rigid, well-learned strategies rather than adapting flexibly to new contingencies (Browning et al., 2015).

R3.14. In page 6, the sentence “At the intervention-level we manipulated task predictability (consistent vs. inconsistent) during training...” A comma is missing after “At the intervention-level”.

Thank you. The missing comma has been added.

R3.15. It would be easier to follow if you list the hypothesis as H1, H2, H3, etc... and in the result section, indicating whether each hypothesis was supported.

We have now included tags for H1-H4 in the Present Study section of Introduction, and refer to H1-H4 throughout results to indicate where evidence supports or fails to support those hypotheses. The Present Study section now reads:

We preregistered several hypotheses, separately for anticipated effects of learning during training (https://osf.io/wj68f/?view_only=2dbf80a1928c411eb7c79daf990269e0), and transfer post-training (https://osf.io/x72z3/?view_only=da1984f9ff244ca798fa6359e1b749b7). At the intervention level, we hypothesized that performance would be greater in the consistent condition than the inconsistent condition (H1), and that performance would be greater in a less

stressful context than a more stressful context (H2). We further hypothesized that changes in amount of practice would interact with training condition (consistent vs. inconsistent) and post-training condition change (e.g., switching from consistent to inconsistent) to predict learning and transfer (H3). Finally, we hypothesized that several emotion-cognition traits at the between-person (i.e., individual differences) level would interact with training condition and stress context at the intervention level, including elements of emotion regulation, intolerance of uncertainty, and behavioral inhibition (H4). Through a series of models, we demonstrate the contributions of each level of analysis (i.e., intervention, between-person, within-person) in understanding individual differences in acquisition and transfer of learned skills, and how those differences deviate from group-level findings.

R3.16. The first sentence in the method section “Half the participants were either recruited from the university’s SONA subject pool, the other half from the local Cleveland community, in exchange for course credit or \$80 USD...” The word “either” should be removed.

Thank you. This sentence now correctly reads:

Half the participants were recruited from the university’s SONA subject pool, the other half from the local Cleveland community...

R3.17. All “ β ” sign should be italicized.

Thank you. All β s are now italicized.

Editor

E1. Your manuscript titled "Task, person, and experiential characteristics drive the transfer of learning" has now been seen by our reviewers, whose comments appear below. In light of their advice I am delighted to say that we are happy, in principle, to publish a suitably revised version in Communications Psychology. We therefore invite you to revise your paper one last time to address the remaining concerns of our reviewers and a list of editorial requests. At the same time we ask that you edit your manuscript to comply with our format requirements and to maximize the accessibility and therefore the impact of your work.

Thank you very much for the opportunity to revise our manuscript. We are pleased that the paper is invited for publication in principle at Communications Psychology and have carefully addressed all remaining reviewer concerns and editorial requests. We have also revised the manuscript to fully comply with the journal's format requirements and to maximize accessibility.

Reviewer 1

R1.1. The authors were incredibly responsive to my previous comments. All of the changes that were made in text fully addressed the given points and the responses to analyses (e.g., functional form) or presentation (e.g., individual difference plots) were reasonable and compelling. I have no further questions or comments. I think the work is really interesting and should make an excellent contribution to the literature.

We appreciate your thoughtful review and are glad that the revisions addressed your earlier concerns. Thank you for helping us strengthen the paper's contribution.

Reviewer 2

R2.1. Thank you for the alterations made in the manuscript. I think it greatly improved the flow the study and the argumentation. Below I provide a number of additional comments.

Thank you for your thoughtful evaluation.

R2.2. In the introduction, you mention that there is a lack of focus on interactions between variables in predicting transfer. I think it would be good to explain a bit more why this is problematic as this argumentation is currently lacking. To be able to substantiate this argumentation, I would recommend the manuscript of Blume et al. (2019) in which an interactionist perspective is also explained including potential contributions of adopting such an approach.

In the revised manuscript, we have expanded the introduction to clarify how ignoring interactions between task, person, and experiential variables can mischaracterize transfer effects

and to explicitly align our approach with an interactionist perspective on training and transfer (e.g., Blume et al., 2019). We have also added Blume et al. (2019) to the reference list.

In the Introduction we now say:

Differing from the longstanding focus on training and cognitive abilities in the learning transfer literature, a recent review (von Bastian et al., 2022) suggests that accounts focusing solely on training quality or on learner traits are incomplete. When models omit interactions between task design, learner characteristics, and experiential histories, they risk understating transfer in subgroups where conditions and traits align. They can also obscure meaningful outcomes when positive and negative conditional effects cancel each other at the group level. An interactionist perspective on transfer emphasizes that the impact of a training intervention depends jointly on who is being trained, how, and for what future context, and highlights the value of modeling these cross-level contingencies explicitly (Blume et al., 2019). Consistent with this view, we frame transfer as the outcome of a dynamic interplay among intervention-level factors, between-person differences, and within-person fluctuations.

R2.3. In the present study section, you do mention some contributions for practice based on your study, but this could be made more concrete. Especially as to how practitioners can use insights of your study for training or work practice.

Thank you for this suggestion. We agree that our practical implications could be made a bit more concrete.

In the Introduction we now say:

This study seeks to advance transfer theory by jointly manipulating task structure and stress while modeling between-person traits and within-person practice. Consistent contexts should foster efficient, model-free routines that transfer poorly to volatile settings; inconsistent contexts should recruit model-based planning, supporting transfer when contingencies change but taxing learners unevenly depending on stress reactivity and emotion-cognition traits. For practitioners, this multi-level account yields potential opportunities for training variability (e.g., switching between consistent and inconsistent environments), stress context intervention (e.g., practicing under stress), and trait-informed scaffolds (e.g., identifying favorable training environments based on individual traits and preferences).

R2.4. It is good that you provide more explanation for using more objective measures in the method section. But I think it is also in itself a valuable contribution to the transfer literature. Perhaps it would be good to add it as a contribution of your study in the theoretical framework.

We agree that our emphasis on objective performance, efficiency, and physiological measures is itself a contribution to the transfer literature, and we now make this more explicit in the theoretical framework.

In the Introduction / theoretical framework section, at the end of the paragraph that lays out our multi-level approach and before the preregistration paragraph, we now say:

Methodologically, this study contributes to the transfer literature by combining objective performance and efficiency measures with physiological indices of stress reactivity, providing a process-oriented view of transfer that goes beyond self-report or coarse achievement outcomes.

R2.5. No further comments regarding the results section.

N/A.

R2.6. In the discussion section, you discuss the need for person-centered analyses in understanding transfer. Perhaps a study done by De Jong et al. (2023) and/or Quesada-Pallares et al. (2022) could be interesting as these study also employed a person-centered approach towards transfer. Especially with regard to how findings in your study relate to their findings.

Thank you for this suggestion. We agree that our emphasis on person-centered and idiographic perspectives aligns with recent person-centered work on transfer, and we now explicitly connect our findings to De Jong et al. (2023) and Quesada-Pallares et al. (2022) in the Discussion.

The heterogeneity in our performance trajectories further underscores the value of idiographic and person-centered methods for designing and evaluating training that is responsive to individual profiles rather than group averages (Beck & Jackson, 2022). While group-level trends revealed general patterns of learning transfer, individual differences in emotion-cognition traits demonstrated substantial heterogeneity in responses. Recent person-centered work on transfer motivation and transfer factors likewise shows that distinct trainee profiles differ systematically in the extent to which training is implemented on the job, underscoring that averaging across subgroups can obscure meaningful patterns in transfer (De Jong et al., 2023; Quesada-Pallares et al., 2022). Our findings parallel this work by showing that subgroups defined by emotion-cognition traits follow different pre-switch and post-switch trajectories. For example, participants with high intolerance of uncertainty struggled more in inconsistent training contexts and demonstrated poorer transfer outcomes, whereas those with low intolerance exhibited stronger pre-switch performance and faster convergence post-switch. These findings support prior calls for methodologies that prioritize within-person variability to capture individual learning and transfer trajectories accurately (Grice et al., 2012; Fisher 2018) and extend them by linking trajectories to emotion-cognition and physiological profiles.

R2.7. In the limitation section, you mention that the age range might limit the generalizability of your findings. Here you could provide some more explanation as to why this limits your findings towards a broader population.

We have expanded the limitation regarding our restricted age range to clarify why it constrains generalizability to broader populations.

Finally, our sampling frame was restricted to younger adults (18–30), which limits generalizability to middle-aged and older adults. Age is associated with systematic changes in autonomic functioning (e.g., heart-rate variability, blood pressure), cognitive control, and emotion regulation, all of which may alter how stress and task structure influence learning and transfer. As a result, the patterns observed here may not fully capture how older adults adapt to changing contexts, and future work could explicitly test age as a moderator of stress reactivity and transfer.

R2.8. You have now more clearly worked out implications for both theory and practice. However, I think that these implications could be described in a more structured manner. In the future direction paragraph, the last alinea is more a practical implication, whereas in the implication paragraph you also mention direction for future research (first alinea). Making a clearer distinction is recommendable.

We have removed the paragraph from the Future Directions section that also discussed implications and was largely redundant with the implications paragraph, resulting in a clearer distinction between future research and implications.

R2.9. I would recommend to end your manuscript with a conclusion paragraph in which you sum up the main contributions of your study.

We have added a brief concluding paragraph that summarizes the main contributions of the study.

By measuring objective performance, efficiency, and autonomic responses, we demonstrate how group-level transfer effects can mask compensatory adaptations between individuals. This multi-level, person-centered approach extends existing transfer theories and offers a framework for designing training environments that are both structurally robust and responsive to individual profiles. Together, our findings suggest that transfer of learning may be jointly shaped by task structure, stress context, and individual differences in emotion-cognition and physiological regulation.

Reviewer 3

R3.1. Thank you for your careful and comprehensive revision. I have no further concerns or suggestions. Congratulations on this valuable contribution, and best wishes for your continued research.

Thank you very much for your thoughtful review and supportive final assessment! We greatly appreciate your time and feedback.